# Triboelectric micro-flexure-sensitive fiber electronics

Shaomei Lin[1], Weifeng Yang[1], Xubin Zhu[1], Yubin Lan[2], Kerui Li [1],
Qinghong Zhang [3], Yaogang Li[3], Chengyi Hou [1] ✉ & Hongzhi Wang [1] ✉

Developing fiber electronics presents a practical approach for establishing multi-node distributed networks within the human body, particularly concerning triboelectric fibers. However, realizing fiber electronics for monitoring micro-physiological activities remains challenging due to the intrinsic variability and subtle amplitude of physiological signals, which differ among individuals and scenarios. Here, we propose a technical approach based on a dynamic stability model of sheath-core fibers, integrating a micro-flexure-sensitive fiber enabled by nanofiber buckling and an ion conduction mechanism. This scheme enhances the accuracy of the signal transmission process, resulting in improved sensitivity (detectable signal at ultra-low curvature of 0.1 mm$^{-1}$; flexure factor >21.8% within a bending range of 10°.) and robustness of fiber under micro flexure. In addition, we also developed a scalable manufacturing process and ensured compatibility with modern weaving techniques. By combining precise micro-curvature detection, micro-flexure-sensitive fibers unlock their full potential for various subtle physiological diagnoses, particularly in monitoring fiber upper limb muscle strength for rehabilitation and training.

A network of body area sensors constructed with fiber electronics is significant in developing technologies that rely on fiber and textile-based systems to monitor human health and track activity[1–4]. Developing low-power and self-powered sensors is crucial for advancing distributed wearable devices[5,6], particularly in self-driven triboelectric strain-sensing devices that are easily integrated and adept at adapting to human motion[7,8]. However, many reported fiber-based sensing mechanisms have focused on enabling high-strain stretchability (e.g., >100%)[9,10], whereas rarely have focused on high sensitivity at micro-flexure deformation[11]. Actually, flexure-sensitive fibers that operate in the latter are more compliant with somatosensory physiology, given the prevalence of such subtle deformations in biological monitoring scenarios[12,13].

Superior to most artificial sensors[14–17], the biological sensory system shows remarkable sensitivity (perceiving deformation within a range of approximately 10 to 100 μm)[18,19], ultra-low threshold

detection (capable of detecting changes as minute as 0.1 mm$^{-1}$)[20], and intelligence (the ability to dynamically adjust its structure and function to response the environment and external stimuli)[21]. These attributes enable organisms to perceive and control subtle deformations, such as pulsations, respiration, heartbeat, and muscle exertion[22,23]. Such extraordinary ability relies on the receptor's structure and the biological mechanism of signal feedback[24–26]. While receptors vary in shape and structure, they share some common structural (i.e., specialized structures like the Meissner corpuscle layered stacking or coiling to amplify stimulus signals) and functional (i.e., ion transport) features[27–30]. These unique structures can enhance the contact area for stimulation, allowing receptors to respond to even the slightest stimuli by deforming appropriately[31,32]. Moreover, the ion conduction mechanism is susceptible to weak electric field changes, enabling it to detect minute potential differences and ensure high-quality signal transmission[33,34]. These characteristics significantly enhance the

[1]State Key Laboratory for Modification of Chemical Fibers and Polymer Materials, College of Materials Science and Engineering, Donghua University, Shanghai 201620, P. R. China. [2]School of Software, Shanghai Jiao Tong University, Shanghai 200240, P. R. China. [3]Engineering Research Center of Advanced Glasses Manufacturing Technology, Ministry of Education, Donghua University, Shanghai 201620, P. R. China. ✉e-mail: hcy@dhu.edu.cn; wanghz@dhu.edu.cn

perceptual capabilities of the receptors, thereby enabling the development of sensors with heightened sensitivity by replicating the receptors' amplification mechanisms.

Inspired by the Meissner corpuscle, we construct a micro-flexure-sensitive fiber by implementing an ordered distribution of nanofiber buckling (NB) at the ionic conductive interface. Meissner corpuscle with cleft space and lamellar cells enables ultrasensitive displacement detection by allowing mechanical compliance. Similarly, the specific bionic strategies in the NB-fiber are as follows: (1) High-specific surface tribo-interfaces are constructed by twisting nanofibers with mismatched modulus ratios on the surface of preprogrammed shape memory hollow fibers. (2) While utilizing annealing to maximize the interfacial gap in situ in a NB-structure. (3) The incorporation of ionogel electrodes enhanced the response to electric field changes, amplified the strength of the stimulus signal, and improved sensitivity (detectable signal at ultra-low curvature of $0.1\,mm^{-1}$; flexure factor >21.8% within a bending range of 10°.) and robustness (cycle number >10,000). Combining flexibility and sensitivity could unlock desirable capabilities in flexure-sensitive fibers, such as muscle force detection cuffs, enabling monitoring of various muscle groups, postures, and force feedback. Consequently, our fiber electronics can significantly improve muscle power control's precision, making it attractive for wearable biomechanical feedback systems.

## Results

### The design principle and continuous manufacture of NB-fiber

Humans possess highly sophisticated sensory capabilities that enable them to perceive and finely control even the most subtle movements and postural adjustments. This extraordinary ability relies on specialized biosensor amplifiers within the human skin[35], exemplified by the Meissner corpuscle—a low-threshold mechanoreceptor (Fig. 1a, left). The Meissner corpuscle is intricately embedded within layered stacks of tissues, featuring a cleft space serving as a mechanical amplifier between the corpuscle and the skin[27]. This amplification enhances the deformation of the nerve endings within the corpuscle, enabling rapid adaptation to stimuli and generating action potentials. Moreover, the bucking structures of the lamellar cells significantly amplify the contact area between the corpuscle and surrounding tissues, further augmenting their ultra-sensitive displacements[25]. We employ a distinct approach to achieve similar effects in pursuit of developing a micro-flexure-sensitive fiber. We enhance the precision of perception by introducing a substantial tribo-gap and a high tribo-interfaces area while encoding the stimulus into an ion signal inspired by the conduction mechanism of action potentials. This sophisticated strategy mitigates signal crosstalk, creating an artificial micro-flexure-sensitive fiber with exceptional sensitivity and an impressively low detection threshold (Fig. 1a, right).

The sensitivity of the NB-fiber is accurately assessed by measuring the equilibrium gap distance (h) between two tribo-interfaces, showing a direct correlation with the voltage signal (Supplementary Fig. 1). We establish an equation to determine the moment when the sheath and core of the coaxial fiber electronics achieve complete contact[36,37].

$$h = \frac{\sqrt{\left(1 - \frac{(r+R+h_f)\theta}{2n^2}\right)^2 - 1}}{n} \qquad (1)$$

where R denotes the radius of the core, $h_f$ indicates the thickness of the sheath, $r$ represents the radius of curvature of the system, $n$ signifies the wavenumber of the sheath, and $\theta$ denotes the flexure angle. With fixed $R$, $h_f$, and $r$ values, the gap distance decreases as the wavenumber increases. Consequently, the flexure sensitivity of the electronic fiber can be finely tuned by modifying the amplitude of wrinkling in the fiber, allowing for precise adjustment of the sensing capabilities (Supplementary Fig. 2 and Supplementary Note 1).

Significantly, developing micro-flexure-sensitive triboelectric fibers within the confined space of one-dimensional fibers (mean diameter <1 mm) presents a challenge. These fibers require high-resolution electrical signal data to amplify minute flexures, typically necessitates significant adjustments in the tribo-gap and contact area to optimize charge transfer during flexure stimuli[38]. By meticulously analyzing the deformation energy levels within the substrate and nanofiber network (Supplementary Fig. 3), we achieved the largest tribo-gap under flexure deformation compared to existing fiber structures (Fig. 1b).

Figure 1c depicts the manufacturing process of NB-fiber. First, the ethylene-vinyl acetate (EVA) shape memory hollow fiber underwent varying degrees of pre-stretching using a tensioner. The electrostatic field is then used to pull the nanofibers to the metal funnel while regulating the rotation speed of the funnel and the collecting roller's speed to twist the tribo-positive thermoplastic polyurethane (TPU) nanofibers onto the surface of the EVA hollow fiber (Supplementary Fig. 4 and Supplementary Movie 1). As a result, the shape memory fibers exhibit some stretchability (~100%, as shown in Supplementary Fig. 5), which enhances the dynamic stability of the buckled nanofiber network. Second, the ionogel solution was introduced into the hollow fibers to construct the electrodes (Supplementary Note 2 and Supplementary Figs. 6 and 7). Finally, the fiber filled with the electrode solution was placed in a heating pipe to release the stress, allowing the in-situ construction of NB-structure and completing the one-step curing of the electrode. Ultimately, a hundred-meter-length (not limited to this size) NB-fiber was continuously fabricated. By carefully adjusting production parameters, we ensure consistent NB-fiber properties even during mass production (Supplementary Figs. 8 and 9).

The large tribo-gap of NB-structure was achieved through an in-situ annealing process using preprogrammed shape memory polymers. Semicrystalline polymers consist of various crystals with different melting points, each of which can remember a temporary shape after programming[39]. When the material is heated to a specific activation temperature, $T_{activation}$, within its melting temperature range, crystals with melting temperatures below this threshold revert to their original shapes due to entropic elasticity. Meanwhile, the temporary shapes held by crystals with melting temperatures above $T_{activation}$ remain frozen. Notably, the slight and gradual increase in $T_{activation}$ ensures smooth and stepless morphing, facilitating the successive restoration of temporary shapes. In our study, the deformation temperature of the shape memory material starts at 80°C, ensuring stability within the typical range of human activities (Supplementary Fig. 10). Additionally, the abundant chain entanglement within the material contributes to the structural stability of the hollow fiber, preventing collapse during activation of the shape memory effect.

Figure 1d presents the statistical analysis of roughness for NB-fiber subjected to various curvature deformations, along with the ultra-deep 3D microscopic surface before and after flexure. Furthermore, Fig. 1e illustrates the corresponding change process in wrinkling amplitude from the initial state to the dynamically stable turning at $521\,\mu m^{-1}$. Notably, the wrinkling amplitude significantly decreases as the system undergoes flexure. The height of wrinkles directly impacts the effective surface area available for contact and separation. Within a specific range of bending curvatures, a more significant variation in wrinkle amplitude leads to substantial changes in the contact-separation distance and effective surface area. This, in turn, results in an enhanced triboelectric voltage response to bending motion. Therefore, it confirms the efficacy of the large wrinkle gap structure in augmenting the fiber's flexure sensitivity.

### Working mechanism and electromechanical performance of the NB-fiber

Due to the ingenious tribo-gap design, our fibers exhibit unparalleled flexure-sensitive properties. The underlying flexure sensing

mechanism is as follows: during the flexure process, the time-varying electric field generated by the tribo-interface induces electrostatic induction, thereby causing electrical signals, as depicted in Fig. 2a. A layer of buckled nanofiber network is attached to the outer wall of the EVA fiber, serving as the tribo-positive material. The ionogel affixed to the inner EVA fiber acts as the electrode for triboelectric fiber. The varying axial flexure strain applied to the EVA shape memory fiber and

the curved nanofiber network induces a consistent change in the tribo-gap between them throughout the flexure-releasing cycle. Consequently, this alteration modifies the sensing potential of the electrode material and facilitates the flow of ions within the ionogel through electrostatic induction. Distinct from metal electrodes, it endows exceptional responsiveness to flexure deformations[40]. They confer numerous benefits regarding (high-frequency) signal anti-

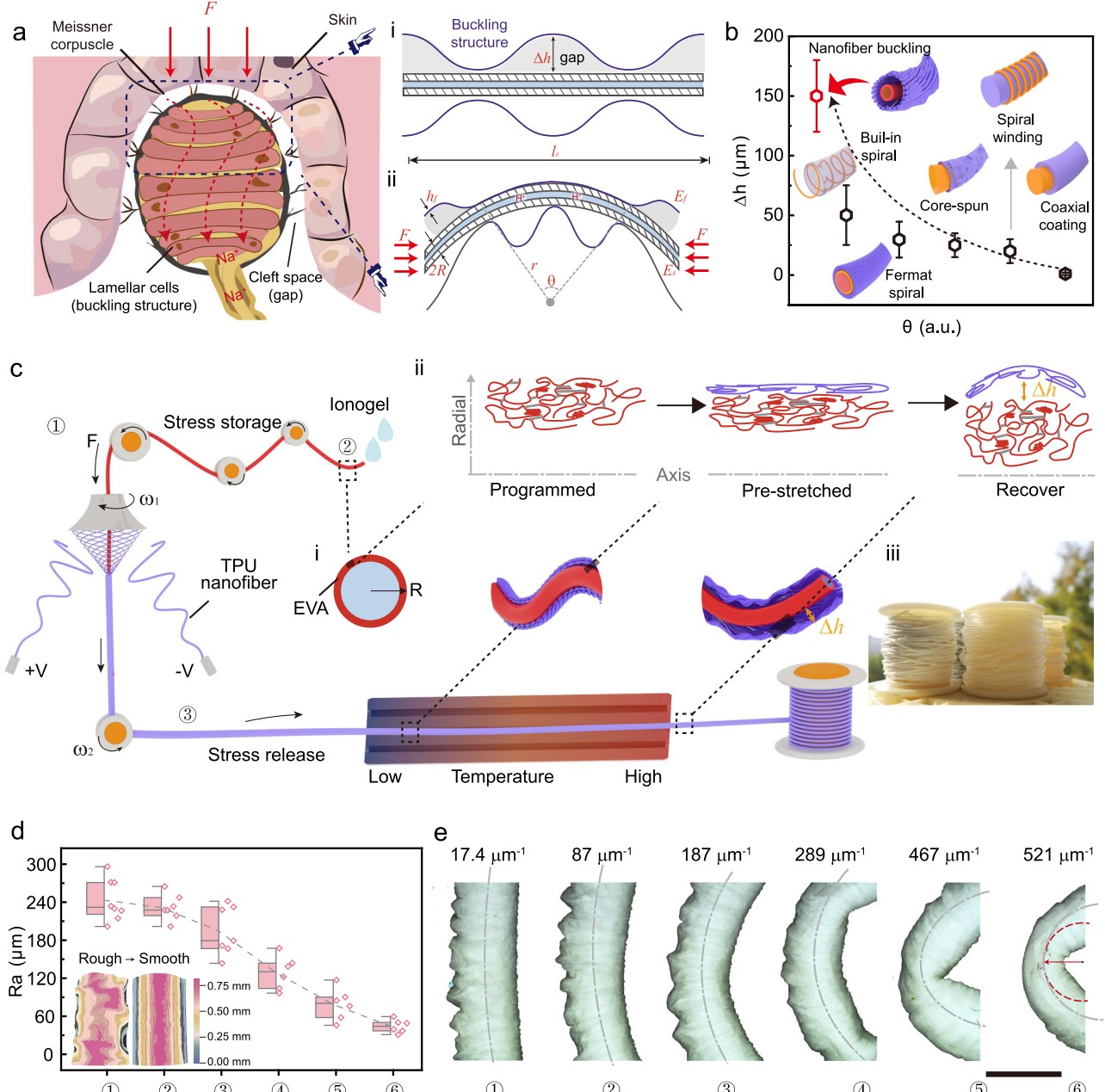

**Fig. 1 | The design principle and continuous manufacture of NB-fiber. a** Left: Illustration depicting the combination of mechanical structure and sensory transduction in biological skin, labeling the amplification and conversion process of the Meissner corpuscle's (sensitive touch) cleft space and lamellar cells to external mechanical stimuli. Right: Schematic diagram illustrating the structure of an e-fibers system for an artificial micro-flexure sensor. The buckling structure and gap magnify external micro-stimuli (pressure and deformation) and enhance feedback accuracy through force-electric conversion. **b** Tribo-gap comparing of state-of-the-art fiber structure designs, including axial coating, spiral winding, core-spun, Fermat spiral, built-in spiral, and NB-structure, under various angular deformations. **c** Schematic illustration of NB-fiber continuous and scalable

manufacture based on conjugate electrospinning. Tensioners regulate the pre-programmed shape memory hollow fibers to achieve desired shapes. The ionogel serves as electrodes with tunable mechanical properties. The change of heating ring employed to induce the formation of the NB-structure from blue to red represents the shift from low-temperature zone to the high-temperature zone. (i) Geometric model of NB-fiber during different preparation steps; (ii) Molecular chain transformation; (iii) Collection of NB-fiber on the reeling roller (scale bar: 2 cm). **d** Ultra-deep 3D microscopy image of NB-fiber with roughness statistics, and **e** wrinkled structure dynamic stability flexure from pristine state to 521 μm⁻¹. (Scale bar: 1 mm).

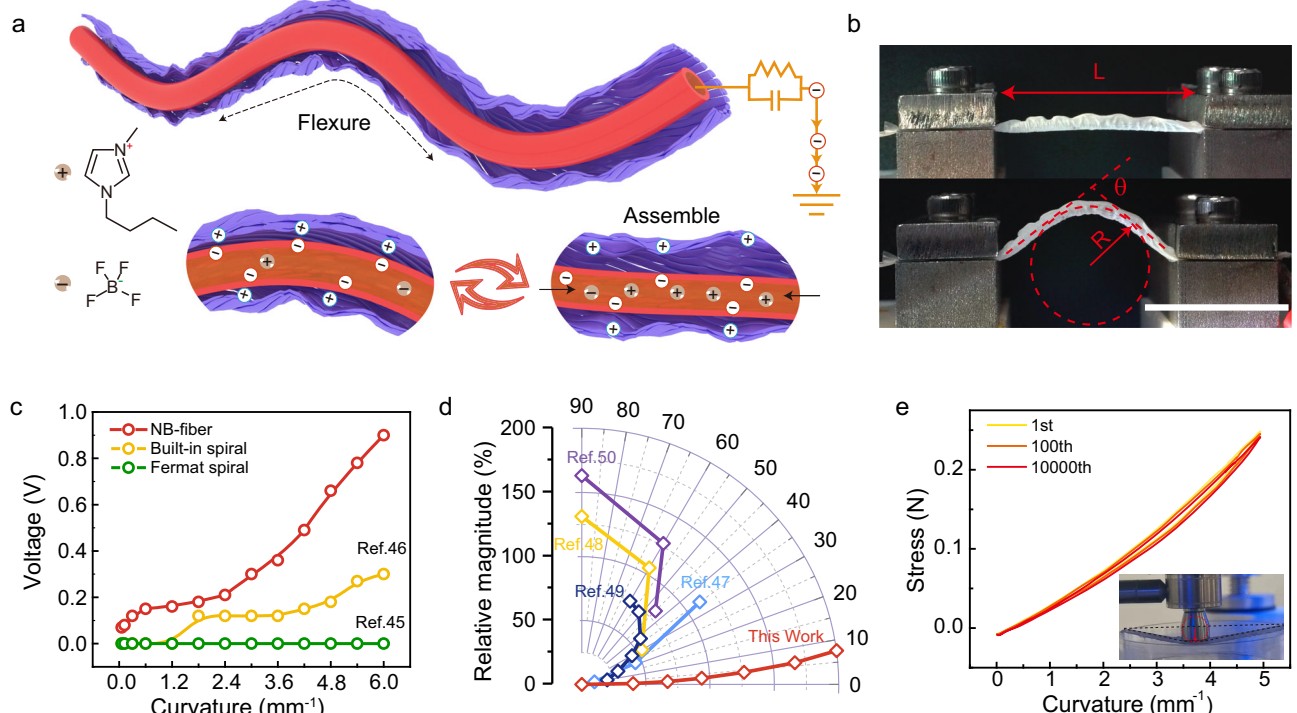

**Fig. 2 | Working mechanism and electromechanical performance of the NB-fiber. a** Triboelectric mechanism of electron-ion conduction in NB-fiber. **b** The images illustrate the test setup of NB-fiber in both the initial and flexure states. scale bar: 2 cm. **c** A comparison of the sensitivity of NB-fiber with various fiber structures. **d** A sensitivity comparison was conducted between the current study and the latest findings reported in the literature, normalizing the rate of change of signal amplitude under identical stimuli. **e** Flexural fatigue testing of NB-fiber.

interference, effectively minimizing signal crosstalk and ensuring the transmission of high-fidelity signals (Supplementary Fig. 11).

To accurately characterize the mechanical and electrical properties of a fiber-based bendable sensing unit, we employed an actuating motor to provide cyclic squeezing force as the input stimulus, as depicted in Fig. 2b. The corresponding test results are presented in Supplementary Fig. 12 and Supplementary Note 3.

State-of-the-art fiber structure designs, such as axial coating[41], spiral winding[42], and core-spun[43,44], typically offer larger specific surface areas by decorating nanostructures to obtain nanoscale roughness. Specifically, the Fermat spiral spinning technology enables the ordered assembly of nanofiber networks[45]. However, its limited stress storage-release capability results in a minimal interfacial gap, insufficient sensitivity, and minor potential change under subtle topological transitions. Thermal drawing offers the potential for continuous construction of a large gap using prefabricated hollow structures[46]. However, the similarity in flexural moduli and restricted topological deformation of inner electrodes limit the significant change of the tribo-gap during flexure. In contrast, the NB-structure formed via the subcritical secondary bifurcation instability of the cylindrical sheath-core configuration exhibits exceptional sensitivity to flexure deformation, even at a curvature radius as low as $0.1\,\text{mm}^{-1}$. Notably, an impressive voltage response of 0.9 V was achieved when the radius of curvature reached $6\,\text{mm}^{-1}$, surpassing the performance of built-in spiral fibers and Fermat spiral fibers (Fig. 2c and Supplementary Note 4). Moreover, in the realm of triboelectric and wearable textiles for biosignal sensing, NB-fiber's flexure factor surpasses that of the majority of flexure sensing devices (Supplementary Table 1). These findings underscore the NB-structure's structural superiority, emphasizing its enhanced ability for flexure sensitivity.

We compared NB-fiber and previously reported bending sensors in terms of normalized sensitivity, representing the sensing signal's relative amplitude change under identical bending angles (Supplementary Tables 2 and 3)[47–50]. The findings reveal that NB-fiber exhibits remarkable responsiveness even at ultra-low angles (0.4°), with a flexure factor exceeding 21.8% within a bending range of 10° (Supplementary Fig. 13 and Supplementary Note 5). This performance surpasses specific active sensing devices such as piezoresistive and capacitive bending sensors (Fig. 2d). The superiority of NB-fiber in detecting and measuring flexure deformations was demonstrated.

To ensure the robustness of compliant and susceptible sensors, the transducing element should resist damage, while the structure should be resilient to rupture or permanent deformation. Supplementary Fig. 14 illustrates the NB-fiber's stable and reversible strain behavior, spanning flexure angles from 0° to 145°. Additionally, the mechanical durability of the NB-fiber is of utmost importance for practical applications requiring prolonged usage. As depicted in Fig. 2e and Supplementary Fig. 15, negligible changes in force were observed for the sensing unit even after 10,000 continuous flexure-release cycles under $5\,\text{mm}^{-1}$. The electrical properties also maintained good stability over repeated cycles (Supplementary Fig. 16). These findings provide compelling evidence that the NB-fiber exhibits high flexure sensitivity, exceptional robustness, and damage resistance.

## Prospects of NB-fiber in subtle physiological diagnosis

The NB-fiber showcases remarkable flexure sensitivity, making it an emerging solution for monitoring human health and delivering medical assistance. Its compact and lightweight design enables convenient attachment to diverse body regions, including the forehead, wrist, chest, and abdomen, facilitating precise measurement of subtle physiological signals. For pulse measurement, a NB-fiber was wrapped around the wrist to enhance deformation during pulsation. Simultaneously, a single NB-fiber was directly sewn onto the arm protector for muscle force monitoring (Supplementary Fig. 17). The schematic in Fig. 3a depicts the sensor's attachment to both the radial artery and the upper arm muscle group. The NB-fiber captures subtle changes in

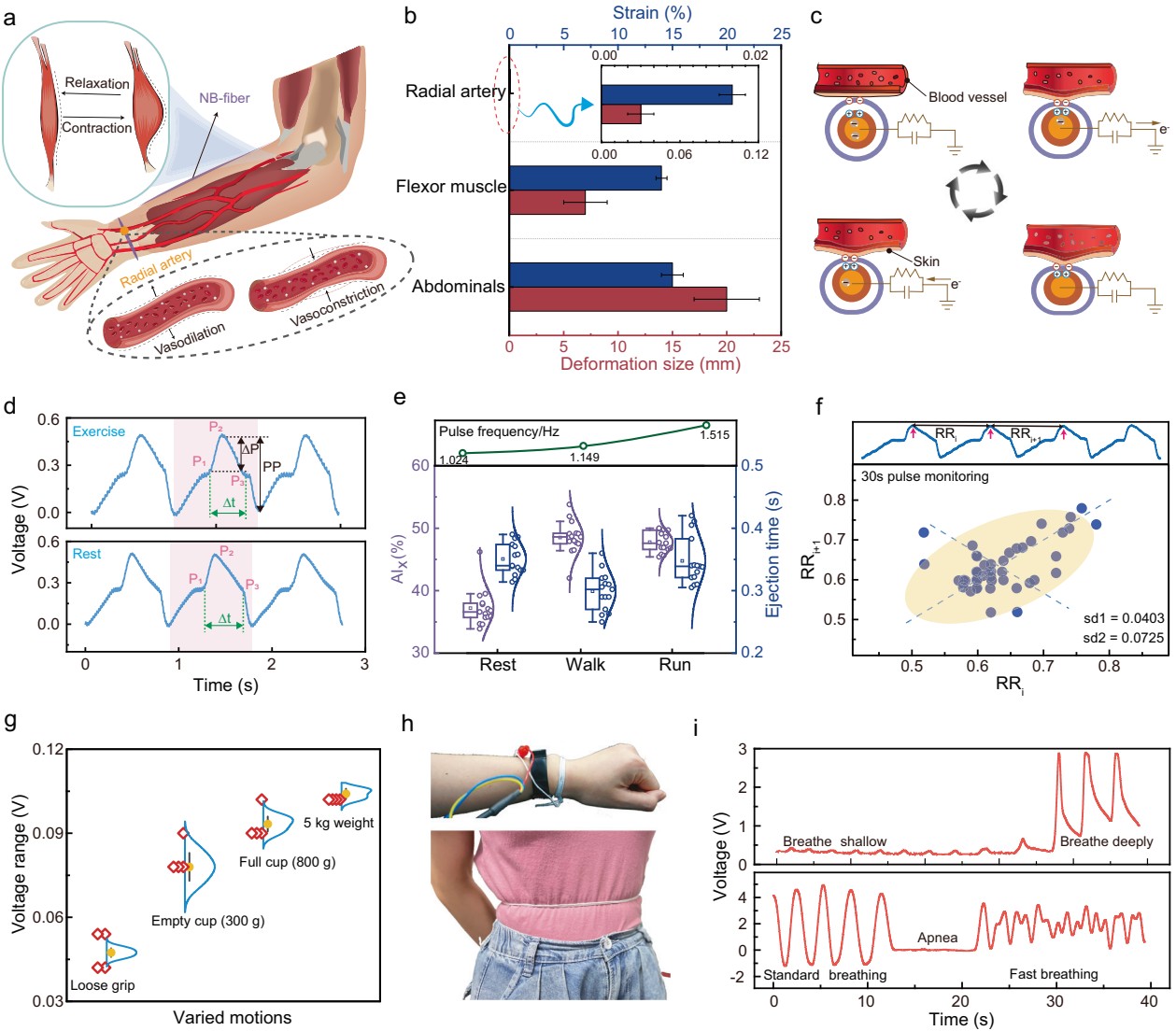

**Fig. 3 | Prospects of NB-fiber in subtle physiological diagnosis. a** The schematic diagram illustrates the placement of the fiber sensor on the arm arteries and muscle groups of the upper extremity, along with the detection principle for monitoring muscle and artery contraction and relaxation. **b** Microscopic physiological activities within the human body are depicted, and the deformation range of various tissues is statistically analyzed. **c** The triboelectric mechanism employed by NB-fiber for pulse detection is presented. **d** Pulse waveforms extracted during exercise and at rest are interpreted and displayed. **e** Various parameters such as Augmentation Index (AIx), ejection time, and pulse waveform frequency are analyzed in different states, including resting, walking, and running. The top panel shows the variation in pulse frequency obtained through Fourier transform analysis. **f** A Poincaré plot demonstrates the pulse detection of a 30-second duration in a 22-year-old male. **g** The electrical signal amplitudes corresponding to muscle strength were measured under different motions. **h** The sensor's schematic diagram for pulse and respiration monitoring was provided. **i** Respiratory waveforms in different states were extracted and interpreted.

curvature arising from the cyclic deformations of blood vessels and muscles, transducing them into discernible voltage variations. Figure 3b comprehensively depicts the deformation range observed within the radial artery, forearm, and abdominal muscles during pulsation, muscle exertion, and respiration. Typically, microscopic physiological phenomena manifest deformations confined within a 5% range[51–53].

Given the delicate nature of each tissue, stringent precision requirements dictate the sensor's accuracy. By integrating flexibility with exceptional sensitivity, the NB-fiber unleashes many desirable capabilities, as illustrated in Fig. 3c. The NB-structure effectively transmits the mechanical deformations of blood vessels, concurrently facilitating ion movement within the NB-fiber electrodes through the triboelectric sensing mechanism. Further scrutiny of the acquired is presented in Fig. 3d, wherein characteristic peaks such as the systolic peak ($P_1$), inflection point ($P_2$), and dicrotic wave ($P_3$) manifest across varying exercise states[54].

Figure 3e exemplifies the NB-fiber's efficacy in discerning human pulse under diverse scenarios, encompassing resting, walking, and running. For instance, a 22-year-old athletic male demonstrates a pulse rate of 1.024 Hz. After a short walk, the pulse rate increases to 1.149 Hz, further escalating to 1.515 Hz during running. The corresponding augmentation index (AIx) and ejection time are presented below, with extensive analysis in Supplementary Fig. 18 and Supplementary Note 6. Lastly, Fig. 3f illustrates the elliptical Poincaré plot of a 22-year-old male athlete, facilitating an assessment of heart rate variability and enabling predictions regarding cardiovascular ailments[55].

Accurate monitoring of upper limb muscle strength is paramount in training and rehabilitation. To address this, we propose the utilization of NB-fiber to monitor the electrical signals generated by muscle groups in various positions within the forearm, as illustrated in Supplementary Fig. 19. Additionally, we tracked the electrical signal amplitudes corresponding to muscle strength, measured under

different motions, as depicted in Fig. 3g, enabling precise evaluation of grip strength. These gripping actions simulate real-world tasks with varying muscle exertion, showcasing their relevance in potential muscle rehabilitation applications (Supplementary Fig. 20). For instance, recovering patients, due to muscle weakness, may only manage a loose grip or lift light objects like an empty cup. Assessing their exertion levels while gripping and lifting different weights provides valuable insights into recovery progress. Moreover, the micro-curvature detection of NB-fiber can aid patients with motor neurological disorders affecting hand and arm movement in achieving motion control of robotic prosthetics, serving as an effective assistive solution (Supplementary Fig. 21).

Breathing, an essential physiological process facilitating the exchange of vital gases, oxygen, and carbon dioxide, is fundamental for sustaining human life. However, respiratory disorders such as sleep apnea syndrome, asthma, and pneumonia can rapidly jeopardize an individual's well-being. Therefore, developing non-invasive sensors for accurate respiratory monitoring is paramount in healthcare management. The NB-fiber, renowned for its ability to capture subtle pulse signals, demonstrates exceptional proficiency in monitoring respiration (Fig. 3h and Supplementary Movie 2). Notably, Fig. 3i showcases the NB-fiber's response signals across a range of respiratory frequencies. The sensor effectively detects breaths with varying rates, encompassing standard and rapid breathing patterns associated with relaxed and exercise states. Control comparisons were conducted at a hospital involving three individuals to evaluate the NB-fiber's accuracy for respiratory monitoring. These tests demonstrated the NB-fiber's high accuracy in measuring key respiratory parameters, such as respiratory rate and tidal volume, compared to measurements conducted on medical equipment (Supplementary Fig. 22).

## Application of NB-textile in biomechanical feedback

The NB-fiber's excellent flexibility makes it compatible with standard textile techniques and equipment, allowing for the creation of infinite-size biomechanical textiles (NB-fiber enabled textile, or NB-textile) when combined with cotton blend fibers (Fig. 4a). The NB-textile inherits the high sensitivity of NB-fiber, enabling spatial mapping of pressure information to gather quantitative tactile feedback information. Furthermore, the textile's properties make the whole system highly flexible and shape-adaptable, allowing it to be stretched, folded, rolled, and bent (Fig. 4b, Supplementary Fig. 23). The wearability and comfort of textiles are of utmost significance[56,57]. To demonstrate the feasibility of potential NB-textile applications, we evaluated the breathable, moisture-permeable, and washable performance of the NB-textile. Supplementary Fig. 24 shows that the air permeability (1380 mm s$^{-1}$) and moisture permeability (0.055 g cm$^{-2}$ h$^{-1}$) of NB-textile far exceed those of TPU non-woven and commercial cotton fabrics, highlighting the controllability of mechanical weaving technology on structural performance. In addition, Supplementary Fig. 25 emphasizes the strong and consistent performance of NB-fiber samples when subjected to different conditions of human skin, whether dry, producing liquid sweat, or oily. To evaluate washing ability, we subjected the NB-textile to ten cycles of washing and drying in a commercial washing machine, according to the rigorous washing test (see "Methods" section and Supplementary Fig. 26). After the washing, NB-textile maintained a stable electrical output under a weight of 10 g (Fig. 4c).

Figure 4d shows the tactile textile array structure of a 6 × 6 pixels array formed by NB-fiber with interlaced latitude and longitude and a signal amplifier connected at the end of each channel. Supplementary Fig. 27 provides an in-depth analysis of the circuitry and functionality of the multi-channel sensing system. The NB-fiber is easily deformed to accommodate bends at the intersection, resulting in a relatively flat contact area between the warp and weft. We used the charge-averaged integrated intensity to represent the touch area to correct for crosstalk due to the capacitance-sensitive characteristics of the textile array.

When a finger is positioned on the sensing array, the NB-textile's curvature at the intersection changes. Notably, both pressing and bending actions induce changes in the curvature of the NB-fiber. Bending involves a distributed force applied across the entire fiber length, leading to curvature changes. Conversely, pressing applies localized force, resulting in significant curvature changes, as depicted in Supplementary Fig. 28-30. The textile array monitors the resulting electrical signal, enabling the detection of both the precise pressure location and the magnitude of the applied load. Figure 4e shows that an unmistakable pressure distribution stimulus signal is generated in the touch area when a finger touches the NB-textile sensor array. In addition, as the index finger moves along the surface of the sensing array (Supplementary Movie 3), the motion trajectory can be dynamically tracked.

To evaluate the potential of NB-textile in rehabilitation training and health diagnosis, we developed a software system for mapping the distribution of human acupoints to capture the characteristic pressure distribution during a massage. To assess the performance of our software system, a comprehensive dataset of users performing various gestures on NB-textile surfaces was recorded (Fig. 4f and Supplementary Movie 4). This dataset, depicted in Supplementary Fig. 31, serves as a benchmark for evaluating the accuracy and precision of our system. In addition, the NB-textile system can monitor muscle conditions during training, enabling rehabilitation physicians to track their patients' progress in real-time and make data-driven decisions about their care, as demonstrated in Fig. 4h and Supplementary Movie 5. Finally, to enhance the usability of our NB-textile system, we have developed an intuitive interface for muscle strength monitoring software, as shown in Fig. 4g. This interface includes the classification of different muscle movements and strength monitoring, enabling clinicians to analyze and interpret the data generated by our system (Supplementary Figs. 32 and 33). Together, these results demonstrate the immense potential of NB-textile as a powerful tool for biomechanical feedback systems, including tactile feedback, action classification, and whole-body human pose prediction.

## Discussion

In conclusion, we have developed an approach to fabricate micro-flexure-sensitive electronic fibers enabled by nanofiber buckling (NB-fiber), which enables the creation of high-specific surface tribo-interfaces and maximizes the tribo-gap, resulting in enhanced sensitivity (detectable signal at ultra-low curvature of 0.1 mm$^{-1}$; flexure factor >21.8% within a bending range of 10°). In addition, our custom software applications, including human acupoint mapping and muscle group force monitoring, demonstrate the versatility of NB-fiber in different fields. For example, it could monitor muscle activity and strength in rehabilitation patients, providing real-time feedback to help optimize treatment outcomes. Likewise, NB-fiber can be used to develop wearable devices to track and analyze movement patterns during physical therapy, thereby providing more personalized and effective treatments. Furthermore, NB-fiber's micro-curvature detection makes it a promising platform for developing advanced prosthetics and exoskeletons, which can significantly improve the quality of life of people with mobility impairments. Overall, NB-fiber has the potential to revolutionize the way we approach healthcare and wellness.

## Methods
### Materials
TPU powder (BASF C65A, Germany); EVA fiber (Zhongshan Xingcheng plastic, China); Dimethylformamide (DMF), Monomers of AAm, AA, the covalent cross-linker MBAA, the ionic liquid solvents 1-Butyl-3-methylimidazolium tetrafluoroborate ionic liquid (BMImBF$_4$), and the thermal initiator ammonium persulfate ((NH$_4$)$_2$S$_2$O$_8$) were purchased from Aladin Chemistry Co., Ltd. and used as received without further purification.

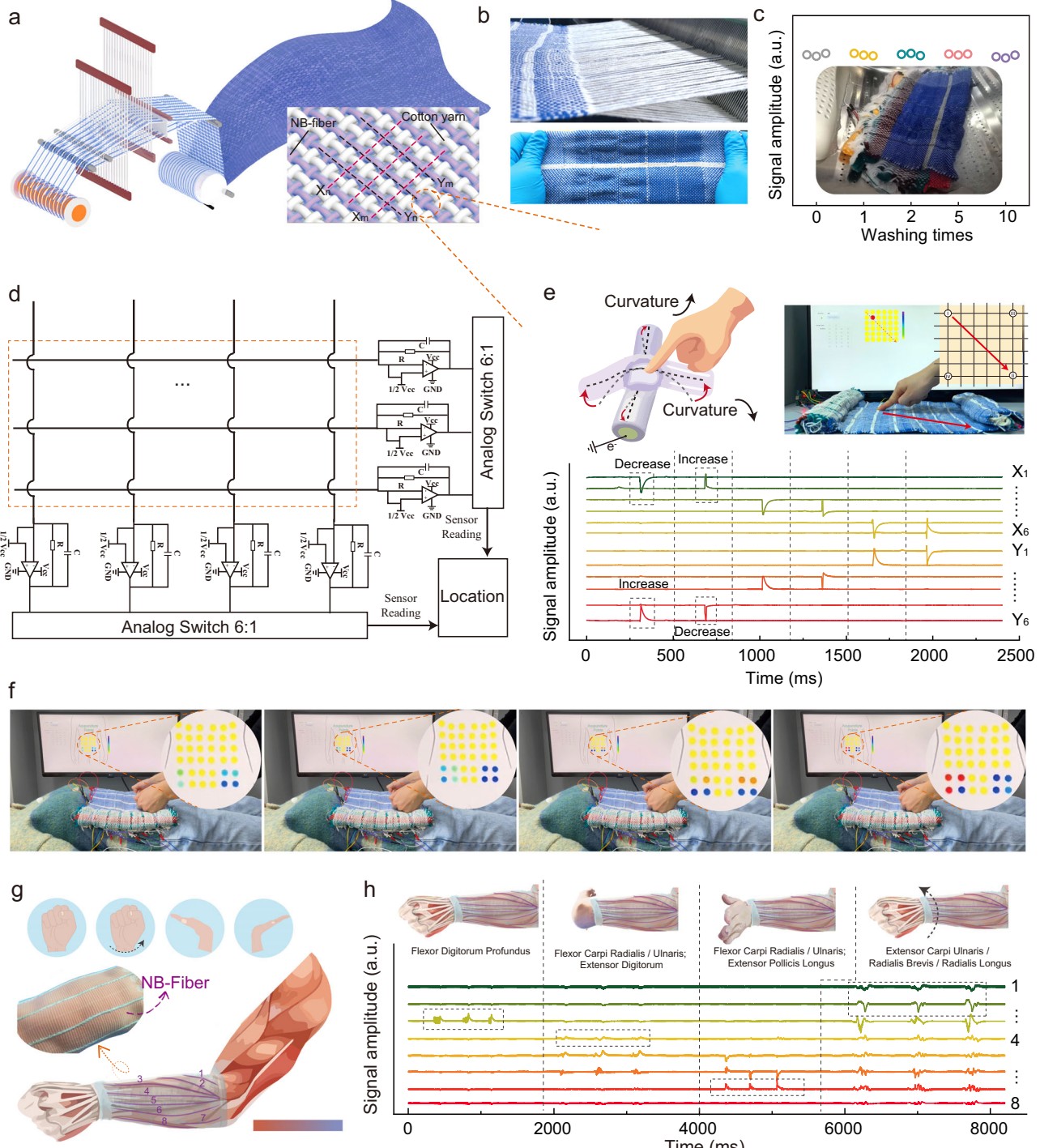

**Fig. 4 | Application of NB-textile in biomechanical feedback. a** The schematic diagram illustrates the weaving process of NB-textile, demonstrating the interlacing technique of alternating NB-fiber and cotton thread as the warp and weft, respectively. **b** A photomicrograph showcases NB-textile woven with a plain weave structure using a commercial loom(top), while a photo of NB-textile under tensile deformation exhibits minimal visible deformation(bottom). **c** Testing washing performance, the inset presents a digital photograph of NB-textile washing using a commercial liquid detergent in a washing machine. **d** A modified isolation circuit architecture was presented for passive sensing array readout. **e** Dynamic path recognition involves the formation of a unit where the sensing warp and weft intersect. Upon applying pressure, the curvature of NB-fiber changes (i). The path process (ii), and the corresponding electrical signal (iii). **f** Pressure distribution during acupressure in humans is demonstrated. **g** The capability of NB-textile to monitor muscle strength is showcased. **h** The strength of muscle groups during different gestures is analyzed.

## Preparation of EVA fiber@ TPU Nanofibers

TPU powder was dissolved in DMF at 60 °C. TPU nanofibers were uniformly twisted on EVA fiber by conjugate electrospinning (TFS-700, Beijing Xinrui Baina Technology Co., Ltd., China) (Fig. 1c and Supplementary Fig. 4). The spinning parameters were as follows: the positive and negative voltages ±9 kV, spinning distance 17.5 cm, propulsion speed 0.03 mL h$^{-1}$, inner needle diameter 0.5 mm.

## Synthesis of ionogel electrodes

See Supplementary Note 2 and Supplementary Fig. 6. Ionogel electrodes were synthesized using a one-step method via random copolymerization of AAm and AA monomers. First, these monomers with a prescribed total concentration ($C_m = 6$ M) and molar fraction of AAm ($x = 0.8$) were dissolved in 3.3 ml BMImBF$_4$ to obtain a homogeneous solution. Then, the covalent cross-linker MBAA ($C_{MBAA} = 0.1$ mol%; in a concentration relative to $C_m$) and thermal initiator ammonium persulfate (0.01 mol%: in a concentration relative to $C_m$) were added. Finally, the solution was poured into EVA fiber @ TPU nanofiber and then hot under 60 for 120 min to obtain the copolymer ionogel.

## Weaving of the NB-textiles

We utilized an automatic rapier loom (SGA 598, Tongyuan Textile Machinery Co., Ltd.) for the continuous weaving of a large-scale blend of NB-fiber with cotton thread to create fabric measuring 0.4 × 1.2 m (Fig. 4b). Digital embroidery: Muscle force detection cuffs' wiring was designed using commercial software (PE-DESIGN 10, Brother) and fabricated with a digital embroidery machine (NV180, Brother).

## Characterization and measurement

Field emission scanning electron microscope (FE-SEM, S-4800, Hitachi, Japan), and ultra-deep field microscope (DVM6, Leica, Germany) were used to characterize the microscopic morphology. The mechanical properties of the fiber were tested by an electronic universal testing machine (Instron 5969 and Electro Force 3230). Keithley 6514 tested the electrical output performance of electronic textiles. Optical photos and demonstration videos were taken using a digital camera (SONY α6300, Japan).

In adherence to ethical standards, the study enlisted five healthy participants aged 20 to 60, including 2 females and 3 males. Internal recruitment within our research group was carried out, and participants received a comprehensive study description that explicitly emphasized the voluntary nature of their involvement, ensuring nonparticipation carried no consequences. The study protocol, titled 'Sensitivity Testing of Fibrous Electronic Devices' with Protocol Number DHUEC-NSFC-2022-43, obtained approval from the ethics review board of Donghua University. Informed consent was secured from all participants, aligning with the testing protocol guidelines. The experiments, which applied sensor fibers on the body surface, involved no stimuli with potential harm to humans.

## Washing test of the NB-textiles

Washing tests followed the ISO 6330 standard for domestic and drying. The electronic textiles underwent laundering in an automatic drum washing machine, following a precise procedure of washing for 15 minutes, rinsing for 10 minutes (3 + 3 + 2 + 2 minutes), and centrifuging for 5 minutes. The inlet water temperature was 40 ± 3 °C. The samples were dried in an A2 condensation-type tumble dryer at a high temperature for 30 minutes, followed by natural cooling after removal.

## Thermal-moisture stability evaluation of the NB-textiles

The air permeability of the textile was tested using an air permeability tester (YG461G, Wenzhou Fangyuan Instrument Co., Ltd., China) following standard GB/T 24218.15-2018. the water vapor transmission rate test was measured by fabric moisture permeability testing apparatus (YG601H, Ningbo Textile Instrument Factory, China) followed by GB/T 12704.1-2009 standard.

## Reporting summary

Further information on research design is available in the Nature Portfolio Reporting Summary linked to this article.

## Data availability

The source data generated in this study has been deposited in Figshare under the accession link (https://doi.org/10.6084/m9.figshare.25187768). Source data are provided with this paper.

## Code availability

The custom code for the wearable biomechanical feedback system used in our demonstration experiments is accessible at https://doi.org/10.5281/zenodo.10625043.

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

## Acknowledgements

This work was supported by the National Natural Science Foundation of China (No.52073057), the DHU Distinguished Young Professor Program (LZA2023001).

## Author contributions

C.Y.H. and H.Z.W. guided the project. H.Z.W., C.Y.H., S.M.L., and W.F.Y. conceived the idea and designed the experiment. S.M.L. fabricated the fiber electronics. S.M.L., W.F.Y., X.B.Z. performed the experiments and measurements. Y.B.L. oversaw the software development. C.Y.H., Y.G.L., Q.H.Z., and K.R.L. revised the manuscript. All authors analyzed the experimental data, drew the figures and prepared the manuscript. All authors discussed the results and reviewed the manuscript.

## Competing interests

The authors declare no competing interests.
