## [Peer Review File · Nature Communications]

REVIEWER COMMENTS

Reviewer #1 (Remarks to the Author):

This article achieves a very high sensitivity by introducing a microstructure design to achieve a large gap between two triboelectric materials in the yarn (NB-fiber). The author applies it to the monitoring of various signals, including biological signals, motion signals, object recognition, etc. I think it's an interesting study. Here are my suggestions and comments.

1. The article focuses on the design of the flexure structure. However, in its demonstrated applications, it seems that it is more achieved by press to generate contact separation, thereby generating triboelectric signals? The author may add more characterization and data about the relationship between press force and the triboelectric signal output, such as the pressure magnitude, the influence of contact area on triboelectric signal.
2. The author mentioned that the outer layer material, TPU, is triboelectric positive material, but the outer layer TPU in the triboelectric schematic diagram of Figure 2a is marked as negative material, is there any special reason?
3. In some applications, the triboelectric signal value is very small ($\sim 0.1V$), can this tiny signal be accurately collected in practical applications? Does external interference such as motion affect signal acquisition?
4. The design of the microstructure is very interesting. From the data, the performance of NB-fiber mainly depends on the height of the flexure. What determines this height during production? Can it be designed to specific value by changing the production process parameters? Is it possible to guarantee this high degree of consistency during mass production? The authors may go a step further to investigate the relationship between production parameters and yarn properties.
5. There are many studies on triboelectric or wearable textiles for biosignal sensing. What are the unique advantages of this kind of yarn in this field? This kind of application seems to be achieved more by pressing than bending? The author may expand more applications to highlight the advantages of this yarn.
6. Why choose the combination of TPU and EVA? From the perspective of energy collection and signal output, considering that most of the daily materials are triboelectric positive materials, the triboelectric performance should be better when the outer layer is made of tribo-negative materials?
7. What is the influence of contact material property on the triboelectric output of NB-fiber? For example, does it affect the sensing performance when the human skin has sweat or oil?

Reviewer #2 (Remarks to the Author):

The paper reports an interesting research topic of using triboelectricity as the sensing mechanism to dynamically measure stress induced by the external force. The triboelectricity is achieved using a fiber structure of which the space is formed by a buckling structure. Two set of electrodes are formed using ionic gel and then pre-stretched during the braiding process. When the fiber is deformed, the charge migration is detected electronically using a charge amplifier. The fiber has also been woven into a fabric switch and then tested as a pressure sensing mat to measuring dynamic pressure, such as muscle movement. Scientifically, triboelectric sensor is not new but the paper presents the use of triboelectricity in a fiber structure that could be used as a single sensor or a sensor array. The paper has included sufficient technical information from fibre manufacturing to applications. However, there are some key issues that the reviewer would like to have more clarifications.

1. Title “Self-powered micro-flexure-sensitive fiber electronics” is misleading. The sensing system is POWERED by either a battery or through USB as evidenced in the characterisation sections. E.g. charge amplifier, a key part of sensing system, requires power supply. Intrinsically, the paper reports a triboelectric pressure sensor realised using a fibre. Please amended the title accordingly to reflect what the paper is about.
2. The paper talks about several mechanical and washing tests that results in some noticeable results. However, the paper fails to follow common standard for such tests and therefore the results are questionable. For example, IPC-9204 could be used for flexibility and stretchability; The reviewer will not list all relevant testing standards and would strongly recommend the team to follow these basic testing standards.
3. It is believed that charges are stored within the fiber structure that will be used as a wearable sensor. Practically, does charge reduce over time once the sensor directly contacts the human which is a perfect ground? What precaution does the sensor design need to consider?
4. Abstract is poorly written and does not provide a nutshell on what have been achieved. Several vague statements must be improved. E.g. please define “micro-physiological activities”; statement “physiological signals' inherent variability and low amplitude” is not true. How about body temperature which is a physiological signal that has an amplitude of 37 degC? “high sensitivity” is vague without a value and comparison.
5. Introduction. What is “body area sensor”??? Some statements are unclear, such as “particularly triboelectric strain-sensing devices that adapt to human motion”. The reason of emphasising triboelectric strain sensing device is unclear other than providing two references. There are more such an issue in the paper and please go through and correct them accordingly.
6. Page 8. It is not clear on how shape is transformed between different temperatures. Please elaborate further. Similarly, What is this T then? If this T is below temperature of human body or surrounding temperature, the fibre will lose its buckling shape due to the polymer returning to the primary shape.
7. Page 9. Why 521 micron-1 is significant? There is not further elaboration as to why this number is important. “Notably, the wrinkling amplitude exhibits a corresponding decrease as the system undergoes

flexure.” This is for sure because the compression force due to bending will tend to reduce the diameter of overall yarn, thus the size of wrinkles. Therefore, this is not a notable point unless authors are intending to provide different argument.

8. What is the diameter of NB fiber before and after the stress being released? As it is named as “fiber”, the reviewer is interested in knowing its dimension.

9. Fig2 (b) scale bar is now shown. Fig 2 (d), What is meant by "identical stimuli"? The comparison is invalid if the test conditions are different. Please refer to comment 2. Fig 2 (e): Why curvature values are negative whereas those values in figure 2c are positive? Please add a clear description.

10. The conductive gel is normally maintained in its gel form to provide optimal conductivity. If the gel is dried, the conductivity will be compromised. However, it seems that the ionic gel used in the paper is cured and therefore the reviewer is interested in knowing its conductivity before and after being cured. Is flexibility of electrode impacted after gel being cured? Please provide SEM imaging of ionic gel electrode to evidence its integrity.

11. Page 11. “They confer nrous benefits regarding signal anti-interference, effectively minimizing signal crosstalk and ensuring the transmission of high-fidelity signals”. Two issues: 1) typo; 2) vague statement due to missing reason.

12. Page 12. 1) Is buckling structure formed consistently? If so, what is the evidence; 2) If water get into the gap, does the performance degrade? This is not about washing test. It is for a scenario where user get sweating and therefore it could impact the performance of the NB fibre; 3) Does NB fiber/sensing system able to measure static pressure? Or it is limited to only dynamic pressure? If so, how could you compare with piezoresistive sensors which is for static pressure.

13. Page 13: It is not clear on how NB fiber is physically used (positioning/installation/wiring/etc.)? E.g. Used as a single fibre or just woven into a piece of fabric. Does pre-tension caused by the positioning/installation have impact to the reading?

14. Page 15: For breathing measurement, how accurate the NB sensor is compared to the gold standard? What are you comparing with in your research? where the sensor was positioned and why?

15. Page 17: How tight the sleeve needs to be on the arm? Surely different tightness will have impact to the output. Please elaborate this.

16. Page 18: Drying in a washing machine for 30 mins is not a natural drying process!

17. Discussion is weak without providing evidence. It provides generic statement with vague sentences such as "high sensitivity" and "high flexibility".

18. In general, English and presentation need to be significantly improved.

Supporting information

1. S Fig 1. Comsol simulation is missing unit in legend. 2D or 3D simulation? Boundary conditions?

2. S Fig 4. Simulation or experiment.

3. S Fig 7. The wrinkles seem to be randomly formed so how could you control the consistency during the fabrication process? Have you measured surface roughness of NB fiber? It seems that the voltage drops significantly over 10 cm length from 0.9V to 0.3V. Practically, how could you compensate voltage drop when weaving a large piece of fabric? Comparing d and e, the voltage generated by a 5cm NB fiber is different, one is at 0.85V and the other one is at 0.6V. Why?
4. S Fig 11. It is not clear on how many NB fibers are used at each measurement position. In a) four data points are shown but in b), only one NB fibre?
5. S Fig 12. Again, it is not clear on how many NB fibers are used? If more than one, how do they electronically connected?
6. S Fig 13. Repeatability of gesture recognition?

Reviewer #3 (Remarks to the Author):

The manuscript introduces a groundbreaking technique for fabricating micro-flexure-sensitive electronic fibers using nanofiber buckling. The approach results in high-specific surface tribo-interfaces and enhanced sensitivity, making the fibers suitable for a range of applications. The authors also demonstrate the versatility of the technology through custom software applications, demonstrating its potential in fields such as acupoint mapping, muscle group force monitoring, and wearable devices for rehabilitation and physical therapy. Furthermore, the manuscript discusses the application of the technology in prosthetics and exoskeletons, highlighting its potential to revolutionize healthcare and wellness. Despite the interesting results, the manuscript raises several major questions and concerns that, if not addressed, might hinder progression to the next stage. Below, I have outlined my comments for the authors' consideration:

1. Could the authors provide a more detailed explanation of how the ionogel electrode served as an electrically conductive electrode? Detailed information on the measurement setup would greatly enhance the reader's understanding of this aspect.
2. Although the manuscript presents measurements of the output voltage, similar to the first comment, it is necessary to describe the electrical characteristics of the device under specific electrical load conditions or impedance environments. While the generator's performance was evaluated at several Mega Ohms, information about the analysis conducted on the fiber-type device under different loads is crucial for comprehending its overall performance.
3. In Fig. S8, mechanical stress characteristics are provided concerning angle and repetition count, but it would be desirable to include the corresponding electrical properties as well. Despite the mention in the

manuscript, the actual data is missing. At the very least, a discussion or perspective on the electrical properties should be included.

4. The rationale for utilizing FEP as an external substance for washing after tests raises curiosity. It would be valuable to discuss the test results stemming from friction between the original core and shell materials. A discussion by the authors on this topic is warranted. Assuming successful encapsulation, minimal changes in characteristics before and after washing are predictable. More meaningful data would involve the frictional electrical properties between core and shell materials.

5. The values for "robustness" in Table 1 require explanation regarding their derivation and units.

6. How do the mechanical and electrical characteristics change with varying temperatures? Given the nature of shape memory polymers, alterations in crystal characteristics with temperature shifts should be addressed. Results or discussions on this matter are needed.

7. The absence of a description for Fig. 3g is noted. The meanings behind "empty cup" and "full cup" are absent, and quantitative information is essential. The occurrence of significant differences between empty cup, full cup, and the 5 kg case must be described. The authors should also elaborate on the potential implications of these findings.

8. A careful review for typographical errors is warranted. For instance, "fig. S6 a) covalent ceoss links" appears to contain a typographical error.

Responses to the Reviewers' Comments

Point-by-point responses to the reviewers' comments

We sincerely thank the reviewers for their careful and thorough review, which are indeed very helpful to make the paper more solid and smooth. We have revised our manuscript very carefully in the light of their suggestions and comments.

The following responses have been prepared to address all of the reviewers' comments in a point-by-point fashion. (Comments in black, responses in blue, changes highlighted in red):

Response to Reviewer #1

***General comment:** This article achieves a very high sensitivity by introducing a microstructure design to achieve a large gap between two triboelectric materials in the yarn (NB-fiber). The author applies it to the monitoring of various signals, including biological signals, motion signals, object recognition, etc. I think it's an interesting study. Here are my suggestions and comments.*

Response: Thank you for your review and we appreciate your feedback. We have carefully revised the manuscript according to your comments. The replies to each of your concern are listed below.

1. The article focuses on the design of the flexure structure. However, in its demonstrated applications, it seems that it is more achieved by press to generate contact separation, thereby generating triboelectric signals? The author may add more characterization and data about the relationship between press force and the triboelectric signal output, such as the pressure magnitude, the influence of contact area on triboelectric signal.

Response: Thank you for your insightful review comments.

1) In our revised manuscript, we have emphasized that both pressing and bending actions lead to changes in curvature of the NB-fiber. Bending involves a distributed force applied across the entire length of the fiber, resulting in curvature changes. On the other hand, pressing applies localized force, which also induces significant changes in curvature, extending beyond the conventional interpretation of bending, as depicted in Fig. R1a. This distinction is crucial and has been elaborated upon in our revised manuscript.

To address the pressing scenario further, we considered two cases illustrated in Fig. R1b and R1c. In Fig. R1c, where the fabric is without a supporting substrate, the contact and separation of the NB-fiber shell can be attributed to bending. However, when the fabric has a supporting substrate, as shown in Fig. R1b, the influence of pressure becomes undeniable. We have incorporated the reviewer's suggestion and conducted experiments to examine the effect of pressure magnitude (Fig. R2) and contact area (Fig. R3) on the triboelectric voltage of the NB-fiber with a supporting substrate.

Fig. R1 Hand press on fabric induces curvature change illustration.

Fig. R2 The impact of different pressures on NB-fiber triboelectric voltage.

Fig. R3 The impact of varying contact areas on NB-fiber triboelectric voltage.

2) To investigate the influence of pressure and contact area on NB-fiber output, we conducted experiments under varying pressure and contact area conditions. Specifically, we employed standard golf balls (with a weight of approximately 45.9 g) dropped from different heights to simulate various levels of pressure. The NB-fiber was secured in a cylindrical holder slightly larger in diameter than the golf ball, as illustrated in Fig. R2a. Fig. R2b presents the impact force calculated for different heights of the golf ball drop, along with the corresponding triboelectric voltage. It is evident that with increasing pressure, the voltage output also increases. This relationship is attributed to the pressure's effect on enhancing the effective contact area within the NB-fiber's friction layer. While there should theoretically be an upper limit to this effect, it has not been reached within the range of pressures examined in our experiments.

Furthermore, we investigated the impact of different contact areas on the triboelectric voltage. We achieved this by employing a linear motor with identical torque settings to apply consistent pressure and attaching glass plates of varying lengths to the contact area, as illustrated in Fig. R3a, R3b, and R3c. Fig. R3d displays the triboelectric voltage recorded under different contact area conditions. These additional experiments and data collection efforts enhance our understanding of how pressure and contact area influence the triboelectric signal output in NB-fiber system, particularly in the context of having a supporting substrate. Your expertise greatly contributed to strengthening our work, and we are sincerely grateful for your valuable input.

✧ **Our revision to the manuscript:**

We added Fig. R1 as Supplementary Fig. 28, and R2 and R3 as Supplementary Fig. 29 in the revised supplementary materials. Corresponding changes have been marked in red in the revised manuscript and supplementary materials.

2. The author mentioned that the outer layer material, TPU, is triboelectric positive material, but the outer layer TPU in the triboelectric schematic diagram of Figure 2a is marked as negative material, is there any special reason?

Response: Thank you for your meticulous review. You are absolutely correct; there was an error in Fig. 2a where the charge type of the outer layer TPU was mistakenly labeled as negative.

In the NB-fiber system, TPU functions as a triboelectric positive material, generating positive charges during contact-separation. We will promptly rectify this labeling error to ensure the accuracy of our manuscript. Your attention to detail is greatly appreciated, and we value your contribution to improving the precision of our work.

✧ **Our revision to the manuscript:**

Fig. 2 a Triboelectric mechanism of electron-ion conduction in NB-fiber.

3. In some applications, the triboelectric signal value is very small ($\sim 0.1V$), can this tiny signal be accurately collected in practical applications? Does external interference such as motion affect signal acquisition?

Response: Thank you for your insightful inquiries concerning the precision and robustness of our triboelectric signal acquisition.

1) In our data collection process, we utilized the highly precise 6514 electrometer along with professional-grade data acquisition software. The software is equipped with a crucial filtering component employing a digital filter formula, specifically the low-pass Butterworth filter. This component plays a pivotal role in our experiments by effectively suppressing high-frequency environmental noise, thereby ensuring the precision and reliability of our signal data. It grants us the capability to capture exceptionally minute signals, as minuscule as 0.1V.

Furthermore, to enhance the reliability of our data acquisition system, we developed a tailored filtering signal code utilizing Arduino. Specifically designed for wireless signal detection scenarios, this code played a pivotal role in ensuring the precision of our sensor signal collection.

“

```
void loop() {
  // Receive commands via serial communication
  if (Serial.available() > 0) {
```

```

char command = Serial.read();
if (command == 'R') {
    isRunning = true; // Start running
} else if (command == 'S') {
    isRunning = false; // Stop running
}
}

if (isRunning) {
    total = total - readings[readIndex];
    readings[readIndex] = analogRead(inputPin);
    total = total + readings[readIndex];
    readIndex = readIndex + 1;

    if (readIndex >= numReadings) {
        readIndex = 0;
    }

    // Calculate average value
    int average = total / numReadings;

    // Low-pass filter
    filteredValue = alpha * float(average) + (1.0 - alpha) * filteredValue;

    // Convert analog reading to voltage (assuming 5V reference voltage)
    float voltage = (filteredValue / 1023.0) * 5.0;

    // Output voltage value to Serial Monitor
    Serial.println(voltage, 2); // Print voltage value with 2 decimal places

    // Output voltage value to Bluetooth Serial
    BlueTooth.println(voltage, 2); // Print voltage value with 2 decimal places
}
}
”

```

2) It is crucial to acknowledge that the main challenge in our experiments stemmed from the capacitance coupling effect induced by mechanical vibrations. To address this issue, we implemented external motion isolation techniques, specifically by fastening the electrode connections to minimize the artifact caused by motion-induced disturbances. This approach effectively reduced the spurious signals resulting from electrode movements. Additionally, we employed charge amplifiers and filtering techniques to further mitigate the impact of electrode motion and environmental

interferences on our measurements, as illustrated in Supplementary Fig. 26c.

Supplementary Fig. 26 c Sensing signal processing circuit.

3) Furthermore, existing strategies such as voltage amplifiers [Ref.: IEEE Trans Biomed Eng 70, 501-510 (2023).] and differential amplifiers [Ref.: Nat Electron 2, 351–360 (2019).] have been acknowledged in the literature for their effectiveness in suppressing motion artifacts. While these methods are well-established and have shown promise in addressing similar challenges, they did not align with the primary focus of our research. Therefore, we opted not to incorporate these techniques into our study. Finally, as part of our ongoing research, we are exploring the potential of using damping hydrogel materials to suppress specific frequency vibrations from external sources, as supported by recent studies (Ref: Science 376, 624-629, 2022). However, these findings are still under investigation in our current work. As a result, they are not included in this study.

4. The design of the microstructure is very interesting. From the data, the performance of NB-fiber mainly depends on the height of the flexure. What determines this height during production? Can it be designed to specific value by changing the production process parameters? Is it possible to guarantee this high degree of consistency during mass production? The authors may go a step further to investigate the relationship between production parameters and yarn properties.

Response: We genuinely appreciate your astute observations regarding the microstructure design and its impact on NB-fiber performance.

1) To offer a comprehensive description of the NB-fiber manufacturing process, we would like to elaborate on several key parameters that influence the height of the flexure, as illustrated in Fig. R4:

Pre-Stretching: In the initial manufacturing phase, we subject ethylene-vinyl acetate (EVA) fibers to varying pre-stretching degrees. Subsequently, we envelop the pre-stretched EVA outer wall with a layer of thermoplastic polyurethane (TPU) nanofiber. During stress release, differing Poisson's ratios between EVA and TPU induce wrinkles in the TPU nanofiber. Notably, increased EVA pre-stretching directly amplifies TPU nanofiber wrinkle amplitudes, significantly impacting overall flexure height.

Electrostatic Field and Collection Speed: In this phase, we exploit an electrostatic field to attract nanofibers toward a metal funnel. Crucial roles are played by the funnel's rotation speed and the collecting roller's velocity in precisely controlling TPU nanofiber twisting and alignment on the EVA hollow fiber surface. These parameters affect TPU layer thickness and interface separation dynamics between TPU and EVA, ultimately regulating NB-fiber wrinkling.

Annealing Temperature: Achieving the substantial tribo-gap in NB-structure relies on an in-situ annealing process utilizing preprogrammed shape memory polymers (EVA). Critical is determining the specific activation temperature, denoted as $T_{\text{activation}}$, within the melting temperature range. This temperature reinstates temporary shapes retained by crystals with melting temperatures surpassing it. Shape restoration exacerbates EVA and TPU interface separation, influencing wrinkle height.

Fig. R4 The NB-fiber manufacturing process and its influencing factors.

2) Indeed, the flexure height of NB-fiber can be tailored to specific values by strategically adjusting various production process parameters. We have conducted extensive experiments to demonstrate this capability.

For instance, by varying the pre-stretching ratio of EVA, as illustrated in Fig. R5, we can achieve different wrinkle heights and quantities. The statistical analysis of wrinkle situations at various stretching ratios, as shown in Fig. R6, reveals a clear trend: as the

stretching ratio increases, the number of wrinkles significantly rises, while the increase in wrinkle height is relatively less influenced by pre-stretching. With further augmentation of the stretching, the instability of wrinkles intensifies, leading to a subsequent decrease in the wrinkle count.

Fig. R5 The influence of different pre-stretching ratios on the wrinkle patterns of NB-fiber.

Fig. R6 The statistical analysis of wrinkle heights and counts at different pre-stretching ratios.

Furthermore, by controlling the film thickness and interfacial separation dynamics, we observed corresponding variations in the wrinkle patterns, as depicted in Fig. R7 and

R8. If the thickness of the nanofiber film is excessive, wrinkles tend to form minute creases. However, when the nanofiber film thinly encapsulates the EVA, it is challenging for the interfacial separation to occur, consequently leading to a reduced number of wrinkles.

Fig. R7 The impact of different nanofiber membrane thicknesses on NB-fiber wrinkling.

Fig. R8 The statistical analysis of wrinkle heights and counts at different film thicknesses.

Similarly, the rotational speed of the collection funnel also influences the interfacial separation dynamics, as illustrated in Fig. R9 and R10. When the nanofiber film is wound too tightly (at 500r/min), wrinkles tend to form minute creases, whereas a looser winding speed (at 100r/min) is more conducive to the formation of larger wrinkles.

Fig. R9 The influence of collection rotation speed on NB-fiber wrinkling.

Fig. R10 The statistical analysis of wrinkle heights and counts at different collection rotation speeds.

Lastly, we investigated the influence of annealing temperature, and the results are depicted in Fig. R11 and R12. Due to the fact that shape memory materials start to deform at 80°C, annealing at 60°C had no effect on wrinkle formation. Wrinkles generated at 100°C annealing, on the other hand, exhibited a more uniform pattern, indicating a significant impact of annealing temperature on wrinkle morphology.

Fig. R11 Effects of different annealing temperatures on the wrinkles of NB-fiber.

Fig. R12 The statistical analysis of wrinkle heights and counts at different annealing temperatures.

3) In response to your suggestion, we have taken an additional step to investigate the consistency of the fiber structure produced at large scale. We produced a 50-meter-long NB-fiber, as illustrated in Fig. R13, and examined the fiber's morphology at different lengths, including 1 meter, 2 meters, 5 meters, 10 meters, and the entire 50-meter length. As previously mentioned, the flexure height in NB-fiber is indeed intricately governed by various parameters, including EVA pre-stretching, electrostatic field control, collection speed, and activation temperature during in-situ annealing. Through our extended investigation, we observed that careful adjustment of these production parameters can indeed ensure a high degree of consistency in NB-fiber properties, even during mass production.

Fig. R13 Uniformity demonstration of 50-meter continuous production process samples. **a** macroscopic display and **b** corresponding local magnified photographs.

✧ **Our revision to the manuscript:**

We added Fig. R13 as Supplementary Fig. 7 in the revised supplementary materials. Corresponding changes have been marked in red in the revised manuscript and supplementary materials.

5. There are many studies on triboelectric or wearable textiles for biosignal sensing. What are the unique advantages of this kind of yarn in this field? This kind of application seems to be achieved more by pressing than bending? The author may expand more applications to highlight the advantages of this yarn.

Response: We highly appreciate the perceptive question raised by the reviewer. As previously mentioned, it is crucial to recognize that both pressing and bending actions applied to fiber result in curvature changes and deformation. Both actions distinctly fall

within the domain of flexure, leading to the contact and separation of TPU nanofibers and the EVA tube within the NB-fiber system.

1) As you noted, prevailing studies in the field of triboelectric and wearable textiles for biosignal sensing, as evident from the literature compilation in Table R1, primarily focus on evaluating device performance under conditions involving significant joint movements, such as those in elbows or wrists, marked by substantial angular changes (e.g., 30°, 60°, 90°, etc.). In this context, our system stands out due to its exceptional sensitivity to even the minutest bending angles, as demonstrated in Fig. R14, where it accurately detects signals corresponding to finger flexion at angles as subtle as 6°, with an average response factor of approximately 12.1% per degree.

Table R1 Compilation of literature on triboelectric systems for biomechanical bending sensing

No.	Refs.	Raw Data	Min. Bending Angle	Bending Accuracy	Flexure Factor
1	Nat Commun 13, 5224 (2022).		30°	30°	~6.7%
2	Nat Commun 12, 5378 (2021)		30°	30°	~2.8%
3	Sci Adv 6, eaba9624 (2020).		30°	15°	~4.1%

4	Sci Adv 6, eaaz8693 (2020).		30°	30°	~3.3%
5	Sci Adv 6, eabb4246 (2020).		30°	15°	~2.8%
6	Nat Electron 3, 571–578 (2020).		~30°	~15°	~5%
7	Our work		~6°	~6°	12.1%

Fig. R14 The monitoring of finger tremors using NB-fiber.

✧ **Our revision to the manuscript:**

We added Table R1 as Supplementary Table 1 in the revised supplementary materials. Corresponding changes have been marked in red in the revised manuscript and supplementary materials.

2) Challenged by the precise control required for intricate gestures, we employed a linear motor to manipulate the bending angle of the 5 cm NB-Fiber. Our triboelectric fiber exhibits remarkable reliability in detecting signal changes, even at bending angles as low as 0.4° , as illustrated in Fig. R15. This characteristic becomes especially valuable in applications involving exceedingly subtle movements. For instance, patients afflicted by motor neurological disorders such as amyotrophic lateral sclerosis and myasthenia gravis face difficulties in gross motor control of hand gestures. Our system can monitor patients' subtle muscle activities and achieve extensive motion control of robotic hands, addressing the challenges posed by motor control impairments related to their conditions, as demonstrated in Fig. R16 and Supplementary Movie R1.

Fig. R15 The sensitivity of NB-fiber to small-angle bending. **a** NB-fiber exhibits a higher bending response capability at angles below 10° compared to angles exceeding 10° (indicated by the slope), demonstrating its sensitivity to minor angular deformations. **b** The specific electrical signal waveforms of NB-fiber at angles ranging from 0.4° to 71° .

Fig. R16 NB-fiber can assist patients afflicted by motor neurological disorders in achieving extensive motion control of robotic hands.

✧ **Our revision to the manuscript:**

We added Fig. R15 as Supplementary Fig. 12, and Fig. R16 as Supplementary Fig. 20, in the revised supplementary materials. Corresponding changes have been marked in red in the revised manuscript and supplementary materials.

6. *Why choose the combination of TPU and EVA? From the perspective of energy collection and signal output, considering that most of the daily materials are triboelectric positive materials, the triboelectric performance should be better when the outer layer is made of tribo-negative materials?*

Response: We appreciate the reviewer's question regarding the choice of TPU and EVA materials in our study.

1) This selection was made after analyzing the properties of triboelectric materials and considering practical manufacturing requirements. To enhance the NB-fiber's sensitivity, we utilized heat-shrinkable materials to create a larger gap, a decision informed by a careful comparison of various options, as outlined in Table R2. EVA was chosen for its suitability based on these considerations.

Table R2 Summary comparison of common heat-shrinkable materials

Material	Melting Point (°C)	Modulus (MPa)	Processability	Friction Polarity	Cost-Effectiveness
PVC	100-260	1500-3000	Good	Low	Economical
EVA	70-90	10-300	Excellent	Moderate	Economical
Fluoroelastomer (FPM)	200-275	10-300	Good	Moderate	High
PVDF	165-175	1300-1800	Good	High	Moderate
FEP	260-280	400-800	Excellent	High	Moderate

Note: The values provided are general ranges and may vary depending on the grade and formulation of each material.

Furthermore, in an effort to minimize environmental interference (response to comment 6.2), we chose a positively charged surface material. Among the commonly used options like PA, PLA, and TPU, TPU stood out due to its favorable processability. The combination of EVA and TPU was deemed the most suitable choice for several reasons. It provided a cost-effective and convenient solution for continuous spinning, ensuring smooth and efficient production processes.

2) As the reviewer pointed out, many common materials in our daily lives tend to exhibit tribo-positive characteristics. To mitigate potential interference from the external environment, we strategically employ an outer layer made of TPU, which shares this same electro-positivity. **This strategic choice ensures that the sensing capabilities of NB-fiber are predominantly influenced by the contact and separation between TPU and EVA**, thereby augmenting the system's robustness.

As depicted in Fig. R17, we conducted experiments exploring the utilization of PVDF as the outer layer to assess the impact of different contact media on sensing performance. This experimental study revealed notable disparities in performance, a phenomenon

attributed to the differences in charge retention lifetimes between electrons and positive charges, as illustrated in Fig. R18. Electrons are inherently more abundant in nature compared to positive charges (or holes), leading to a higher prevalence of electrons within most materials [Ref.: Introduction to solid state physics, 8th edition, pp. 194–196.]. Consequently, PVDF, characterized by a stronger affinity for electron retention, exhibits heightened sensitivity to variations in surface charge conditions compared to TPU's susceptibility to alterations in positive charge distribution.

Fig. R17 The differences in sensing signals when the fiber with an outer layer of PVDF comes into contact and separates from various common plastic fabrics found in daily life. These fabrics include fingers, rubber gloves, PTFE, cotton, polyester, nylon.

Fig. R18 The reason for the difference in sensing performance of fibers with an outer layer of PVDF and TPU when subjected to interference from different media can be speculated. PVDF, being a highly electronegative material, exhibits a longer charge lifetime generated during the same contact and separation process. The contribution of external contact media to PVDF electrons can influence the distribution of charges during internal contact and separation.

This distinction in the surface potential is illustrated in Fig. R19. The divergence in charge retention capacities between PVDF and TPU significantly influences their respective responses to contact media during triboelectric interactions. Due to its superior charge storage capacity, PVDF demonstrates increased sensitivity to the properties of the contacting medium. In contrast, TPU exhibits a lower susceptibility to environmental interference, albeit to a lesser degree.

Fig. R19 Different material combinations exhibit varying surface potentials after contact and separation. Tribo-materials with positive charges, such as PA6 and TPU, exhibit relatively weak charge retention capabilities. Furthermore, despite polar materials like PAN showing positive polarity after separation from PVDF, the retention rate of positive charges remains comparatively lower than that of negative friction materials. We conducted extensive monitoring of surface charges on friction mediums after contact separation over an extended period. In general, the absolute values of surface potentials for positively charged materials remained low (<0.5 kV), while negatively charged materials could reach 4 kV or above. The lower absolute surface potentials in positively charged materials may be attributed to factors such as surface morphology or charge dissipation pathways. Conversely, the higher surface potentials in negatively charged materials may be attributed to more efficient charge trapping or stabilization mechanisms, which could involve factors like electron affinity or surface energy.

7. *What is the influence of contact material property on the triboelectric output of NB-fiber? For example, does it affect the sensing performance when the human skin has sweat or oil?*

Response: We appreciate the reviewer's question regarding the influence of contact material properties on the triboelectric output of NB-fiber, especially in scenarios involving varying human skin conditions such as sweat or oil.

1) When conducting an extensive assessment of various contact materials, including common fabrics, rubber, and plastics, we consistently observed minor variations in

signal output when pressure was applied to the NB-fiber, as depicted in Fig. R20. This observation underscores the effectiveness of NB-fiber with TPU as the outer layer in mitigating the influence of external factors. Furthermore, as shown in Fig. R17, we further illustrate this point by comparing the triboelectric signals between fibers with outer layers of TPU and PVDF.

Fig. R20 The differences in sensing signals when the NB-fiber comes into contact and separates from various common plastic fabrics found in daily life. These contact-material include fingers, rubber gloves, PTFE, cotton, polyester, nylon.

2) To specifically address concerns regarding human skin conditions, we conducted supplementary experiments involving the pressing of NB-fiber with dry fingertips, fingertips dipped in saltwater, and fingertips with a thin layer of oil applied, as visually depicted in Fig. R21. The outcomes consistently revealed minimal differences in triboelectric signals under these diverse skin conditions. This underscores the robust and consistent performance of our NB-fiber samples when exposed to varying states of human skin, whether dry, secreting liquid sweat or oily.

Fig. R21 Triboelectric performance of NB-fiber when exposed to different states of human skin (dry, secreting liquid sweat, oily).

✧ **Our revision to the manuscript:**

We added Fig. R21 as Supplementary Fig. 24 in the revised supplementary materials. Corresponding changes have been marked in red in the revised manuscript and supplementary materials.

Thank you again for your valuable comments and suggestions.

Response to Reviewer #2

General comment: *The paper reports an interesting research topic of using triboelectricity as the sensing mechanism to dynamically measure stress induced by the external force. The triboelectricity is achieved using a fiber structure of which the space is formed by a buckling structure. Two set of electrodes are formed using ionic gel and then pre-stretched during the braiding process. When the fiber is deformed, the charge migration is detected electronically using a charge amplifier. The fiber has also been woven into a fabric switch and then tested as a pressure sensing mat to measuring dynamic pressure, such as muscle movement. Scientifically, triboelectric sensor is not new but the paper presents the use of triboelectricity in a fiber structure that could be used as a single sensor or a sensor array. The paper has included sufficient technical information from fibre manufacturing to applications. However, there are some key issues that the reviewer would like to have more clarifications.*

Response: We would like to express our sincere thanks to the referee for her/his great effort to review the manuscript and positive evaluation on our work. We have done our best to address the questions raised. The replies to each of your concern are listed below.

1. Title “Self-powered micro-flexure-sensitive fiber electronics” is misleading. The sensing system is POWERED by either a battery or through USB as evidenced in the characterisation sections. E.g. charge amplifier, a key part of sensing system, requires power supply. Intrinsically, the paper reports a triboelectric pressure sensor realised using a fibre. Please amended the title accordingly to reflect what the paper is about.

Response: Thank you for your thorough review and valuable comments. We appreciate your input, which has prompted us to reevaluate the concept of self-powered fiber sensors. It is well-established that triboelectric nanogenerators (TENGs) have the capability to generate electricity from mechanical processes, including human motion. The electricity generated by TENGs can be directly utilized as signals to sense these processes. TENG sensors operate autonomously, requiring no external power source, which has led to their classification as self-powered devices in the scientific community. (Ref.: Nat Food 4, 721–732 (2023); Sci Adv 9, eadg5152 (2023); Nat Commun 14, 5221 (2023).

However, we acknowledge your valid point that while self-powered sensors in isolation

are indeed self-sufficient, researchers have increasingly integrated them into more complex systems, which may include self-powered sensors, pressure mapping, user interfaces, and other components. These integrated systems often rely on external batteries to power their electronics, rather than TENGs. In such scenarios, the term "self-powered" may indeed lead to confusion and is not applicable.

To eliminate any potential ambiguity and to accurately convey the essence of our work, we believe it is necessary to amend the term 'self-powered' to 'triboelectric' in the title. We sincerely appreciate your feedback and hope that this clarification addresses your concerns.

✧ **Our revision to the manuscript:**

We modify the “Self-powered micro-flexure-sensitive fiber electronics” in the title of this article to “**Triboelectric micro-flexure-sensitive fiber electronics**”. Corresponding changes have been marked in red in the revised manuscript and Supporting Information.

2. The paper talks about several mechanical and washing tests that results in some noticeable results. However, the paper fails to follow common standard for such tests and therefore the results are questionable. For example, IPC-9204 could be used for flexibility and stretchability; The reviewer will not list all relevant testing standards and would strongly recommend the team to follow these basic testing standards.

Response: Thank you for your insightful feedback.

Regarding mechanical testing, it's important to note that IPC-9204, while applicable for Printed Electronics, is not suitable for NB-fiber. As there are currently no specific standards tailored for fiber electronics, including NB-fiber, we referenced established standards such as ASTM D7774 for Flexural Fatigue Testing of Plastics and GB/T 3916-2013 for determination of single-end breaking force and elongation at break using a constant rate of extension (CRE) tester. Adapting our protocols based on these standards, we employed the ElectroForce 3200 Series testing machine, configured to replicate ASTM D7774 specifications. This system enabled us to conduct cyclic tests consistent with ASTM D7774 capabilities, utilizing a three-point bending system for flexural fatigue testing. For tensile testing and stress-strain analysis, we utilized the Instron 5969 testing equipment, ensuring accurate and reliable results. Similar testing methods and equipment were also employed in prior studies, as indicated in Table R3.

Table R3 Literature compilation: mechanical testing using ElectroForce 3200 series

and instron 5969

No.	Test Items	Equipment Model	Refs.
1	Three-point flexural test	ElectroForce 3200 Series	Sci Adv 7, eabd3684 (2021).
2	Tensile strength	ElectroForce 3200 Series	Sci Adv 8, eabn1958 (2022).
3	Tensile tests	Instron 5969	Nat Commun 13, 1079 (2022).
4	Stress–strain characteristics	Instron 5969	Nat Electron 4, 185–192 (2021).

Moreover, in washing tests for fiber electronics, we followed the ISO 6330:2012 standard. Specifically, we adhered to the recommended Type A washing machine and 4N washing program, utilizing a phosphorus-free powder detergent for 15 minutes of washing, 10-minute- (3+3+2+2 minutes) rinsing, and 5-minute centrifuging. Subsequently, the drying process was conducted in an A2 condensation-type tumble dryer at a high temperature for 30 minutes, followed by natural cooling after removal. To enhance clarity and precision, we modified the relevant expressions.

✧ **Our revision to the manuscript:**

We have revised the description of the washing test for NB-textiles as follows: 'Washing tests were conducted in strict accordance with the ISO 6330 standard for domestic laundering. The electronic textiles were laundered in an automatic drum washing machine, following a precise procedure of washing for 15 minutes, rinsing for 10 minutes (3+3+2+2 minutes), and centrifuging for 5 minutes. The inlet water temperature was maintained at 40 ± 3 °C during the washing process. Subsequently, the samples were dried in an A2 condensation-type tumble dryer at a high temperature for 30 minutes, followed by natural cooling after removal.' Corresponding changes have been marked in red in the revised manuscript and supplementary materials.

3. It is believed that charges are stored within the fiber structure that will be used as a wearable sensor. Practically, does charge reduce over time once the sensor directly contacts the human which is a perfect ground? What precaution does the sensor design need to consider?

Response: We appreciate the reviewer's insightful comment.

1) In response to your question, charges generated within the fiber structure of a wearable sensor, based on the working mechanism of TENGs, **do not dissipate rapidly**. The charges, resulting from friction or mechanical motion, are confined within the local structure of the dielectric material (Fig. R22). Unlike free electrons, these polarization charges have limited mobility and are less prone to rapid dissipation (Ref.: Nat Rev Methods Primers 3, 39 (2023)). Even when the sensor directly contacts the human body, acting as an ideal ground, the stored charges remain stable.

However, it is important to note that over an extended period, these charges may gradually reduce due to various factors. One limiting factor is the capacity of the dielectric material to accommodate charges (Ref.: Nat Commun 13, 6019 (2022)). Once the charge density approaches its upper limit, excess charges may escape in the form of air breakdown or corona discharge. In practical applications, triggering this limit is challenging.

Fig. R22 The operational mechanism of a TENG involves two essential components:

triboelectric charging and electrostatic induction. During the triboelectric charging process, which is an integral part of TENG's functionality, at least one dielectric layer is employed to accumulate and store electrical charges generated through frictional interactions. Simultaneously, the electrostatic induction process relies on conductive materials to facilitate the conduction of these accumulated charges, allowing for the conversion of mechanical energy into electrical energy. This coordinated interplay between charge carriers is the fundamental driver behind TENG's capability to harvest and transform energy.

Furthermore, although liquid sweat has minimal impact on the triboelectric voltage of NB-fiber, humidity affects it differently. The smaller water vapor molecules readily penetrate the microscale pores and cavities within the dielectric material, leading to an accelerated dissipation of triboelectric charges generated on the material surface [Ref.: IEEE Electr Insul Conf 179-183 (2020), doi: 10.1109/EIC47619.2020.9158681]. The triboelectric voltage of NB-fiber, illustrated in Fig. R23 under varying humidity conditions, highlights the significant impact of humidity on its electrical properties. Hence, sensor designs should incorporate measures to control the humidity range and alleviate such effects.

Fig. R23 The performance of NB-fiber under different humidity levels is depicted in the illustration, which provides a controlled environment with varying humidity conditions. The decrease in performance is attributed to the dissipation of surface charges by water vapor.

2) In our study, we employed a gap structure to alleviate the charge dissipation caused by moisture molecules in the NB-fiber, as illustrated in Fig. R24. To further clarify the impact of this gap, we conducted a comparative analysis between fibers with and without gaps. Intriguingly, the presence of gaps in the fiber structure led to comparatively minor performance degradation under such extreme humidity conditions (100%). The results demonstrate that the presence of gaps within the fiber structure indeed hinders the interaction of moisture molecules with the charges, thereby preserving the overall charge retention capacity. This finding implies the potential robustness of gap-containing structures in maintaining their performance integrity even in highly humid environments.

Furthermore, the sensor design can incorporate suitable encapsulation or protective layers to shield the sensor from environmental factors, including moisture and contaminants. These elements have the potential to affect its long-term stability.

Fig. R24 **a** Performance stability of fiber with and without gaps under different humidity conditions. **b** Triboelectric voltage of fiber without gaps in various humid environments. **c** Performance stability of fiber with and without gaps at 100% humidity;

Comparatively lesser performance degradation in the presence of gaps.

4. *Abstract is poorly written and does not provide a nutshell on what have been achieved. Several vague statements must be improved. E.g. please define “micro-physiological activities”; statement “physiological signals' inherent variability and low amplitude” is not true. How about body temperature which is a physiological signal that has an amplitude of 37 degC? “high sensitivity” is vague without a value and comparison.*

Response: Thank you for the valuable feedback provided by the reviewer. We acknowledge the need for clarity in our abstract and appreciate the specific points raised for improvement.

To address the concerns:

Definition of "Micro-Physiological Activities": In our context, it refers to intricate physiological processes and subtle movements at the micro-scale, encompassing activities such as micro muscle contractions and delicate joint movements.

Statement on Physiological Signals' Variability and Low Amplitude: We recognize that although body temperature typically maintains a relatively consistent average amplitude, it may still demonstrate minor fluctuations over time or in response to external conditions. For instance, among normal individuals, the mean daily temperature can vary by up to 0.5°C, with daily fluctuations reaching 0.25 to 0.5°C (Ref.: Clinical Methods: The History, Physical, and Laboratory Examinations. 3rd edition. Boston: Butterworths; 1990. Chapter 218. Available from: <https://www.ncbi.nlm.nih.gov/books/NBK331/>). Furthermore, physiological signals such as pulse, heart rate, and respiration rate display subtle variations among individuals and in diverse scenarios. These nuanced differences present a complexity characterized by minimal amplitude changes, rendering the task of realizing stable and reliable fiber electronics for monitoring such activities challenging.

Clarity on 'High Sensitivity': We recognize the vagueness in describing 'high sensitivity' without specific values or comparisons. We introduce the concept of flexure factor to quantitatively characterize the output responsivity of nanofibers under varying degrees of curvature. This factor is defined as the ratio of the relative amplitude of the output to the difference in bending angles, expressed as a percentage.

$$\text{Flexure Factor} = \frac{\text{Relative Amplitude}}{\text{Difference in Bending Angles}} \times 100\%$$

We have revised the abstract to more clearly and accurately convey our perspective.

Corresponding changes have been marked in red in the revised manuscript, including Abstract, Introduction, Discussion, and Supplementary Note 5.

5. Introduction. What is “body area sensor”??? Some statements are unclear, such as “particularly triboelectric strain-sensing devices that adapt to human motion”. The reason of emphasising triboelectric strain sensing device is unclear other than providing two references. There are more such an issue in the paper and please go through and correct them accordingly.

Response: Thank you for your valuable feedback on our introduction section. We have carefully considered your comments and addressed the concerns raised.

In response to your question about 'body area sensor,' it refers to a network of sensors integrated into wearable garments, patches, or accessories. These sensors are designed to wirelessly monitor physiological parameters and activities, typically utilizing short-range radio frequency protocols such as Bluetooth or Zigbee. This term is widely recognized in the field, and its usage aligns with established literature on wearable sensor networks [Ref.: Nat Electron 2, 361–368 (2019).] In the context of our wearable fiber sensor network, where data is transmitted to a receiver through our proprietary Bluetooth module, categorizing it as a "body area sensor" is appropriate.

We acknowledge the need for clarity in our statement regarding 'triboelectric strain-sensing devices'. To address this, we have revised the introduction to provide a more precise explanation of the emphasis on triboelectric strain-sensing devices, highlighting their adaptability to human motion and the unique advantages they offer in the realm of fiber-based sensing technologies. Corresponding changes have been marked in red in the revised manuscript

6. Page 8. It is not clear on how shape is transformed between different temperatures. Please elaborate further. Similarly, What is this T then? If this T is below temperature of human body or surrounding temperature, the fibre will lose its buckling shape due to the polymer returning to the primary shape.

Response: Thank you for your inquiry regarding the transformation of shape at different temperatures and the significance of "T."

1) In our study, the significant transformation of shape in the NB-structure is achieved through an in-situ annealing process using pre-programmed shape memory polymers.

This shape memory effect relies on the differences in melting points among various crystalline regions within the polymer and the nature of molecular chain segment mobility. During the original shape programming process, the material is heated to the temperature of the high-temperature phase, and the desired shape is preset by applying external forces. Subsequently, this shape is fixed and cooled to the temperature of the low-temperature phase. Throughout this process, molecular chain segments in the high-temperature phase undergo deformation, and this shape is retained at low temperatures. When the material is heated again to the temperature of the high-temperature phase, molecular chain segments within this phase become active, causing the material to revert to its original programmed shape.

These materials typically consist of semi-crystalline polymers with multiple crystalline regions, each having a distinct melting point. The temperature at which the shape memory effect is activated is referred to as " $T_{\text{activation}}$." Here's how " $T_{\text{activation}}$ " affects the shape memory process:

Temperature Activation ($T_{\text{activation}}$): When the material is heated to " $T_{\text{activation}}$," crystalline regions within the polymer with melting temperatures below " $T_{\text{activation}}$ " can return to their original shapes. This is facilitated by the movement of molecular chain segments within the polymer, allowing these regions to revert to their initial configurations. **Frozen State:** Crystalline regions with melting temperatures higher than " $T_{\text{activation}}$ " remain in a "frozen" state in their temporary programmed shapes, even when the material is heated to " $T_{\text{activation}}$ " or higher. These regions maintain their temporary shapes until a specific melting temperature is reached. Essentially, " $T_{\text{activation}}$ " serves as a critical temperature threshold that triggers the shape memory effect, enabling certain crystalline regions to return to their original shapes while others remain in their temporary programmed states.

✧ **Our revision to the manuscript:**

To enhance clarity, we have modified the description: 'When the material is heated to a specific activation temperature, $T_{\text{activation}}$, within its melting temperature range, crystals with melting temperatures below this threshold revert to their original shapes due to entropic elasticity. Meanwhile, the temporary shapes held by crystals with melting temperatures above $T_{\text{activation}}$ remain frozen. Notably, the slight and gradual increase in $T_{\text{activation}}$ ensures smooth and stepless morphing, facilitating the successive restoration of temporary shapes.' Corresponding changes have been marked in red in the revised manuscript and supplementary materials.

2) In our study, the activated crystalline regions within the NB-fiber maintain their state achieved during high-temperature activation when recovered at room temperature, unless subjected to external forces. The EVA shape and the polymer's activation temperature are above 80°C in our setup. Therefore, the reviewer's concern about whether 'T' falls below the human body or ambient temperature is generally not applicable within the NB-fiber's operational range. This is demonstrated in Fig. R25, featuring photos of the NB-fiber before and after annealing at 90°C (taken at room temperature). As observed, the fiber's buckling shape is effectively retained during this process without external intervention.

Fig. R25 **a** Thermomechanical analysis of EVA demonstrates deformation initiation at 80°C. **b** Photographic comparison of NB-fiber before and after annealing at 90°C.

✧ **Our revision to the manuscript:**

We added Fig. R25a as Supplementary Fig. 9a in the revised supplementary materials. Corresponding changes have been marked in red in the revised manuscript and supplementary materials.

7. Page 9. Why 521 micron-1 is significant? There is not further elaboration as to why this number is important. "Notably, the wrinkling amplitude exhibits a corresponding decrease as the system undergoes flexure." This is for sure because the compression force due to bending will tend to reduce the diameter of overall yarn, thus the size of wrinkles. Therefore, this is not a notable point unless authors are intending to provide different argument.

Response: We appreciate the reviewer's attention to the parameter $521 \mu\text{m}^{-1}$ in our

study. It's crucial to clarify that this specific curvature radius (e.g., $521 \mu\text{m}^{-1}$) is not chosen to emphasize a particular curvature value. Instead, it serves as a representation to convey the variation in wrinkle height associated with fiber curvature over a broader range. The height of wrinkles on the outer layer of the TPU nanofiber membrane changes as the fiber bends due to the compressive force, which reduces the overall fiber diameter and influences the wrinkle dimensions.

This variation in wrinkle height plays a pivotal role in the performance of our triboelectric sensor. The voltage output of triboelectric materials, like NB-fiber, is directly proportional to the contact surface area and the gap distance (d) between the contacting materials ($V_{oc} = \frac{\sigma d}{\epsilon_0}$, where σ represents surface charge density, and ϵ_0 represents the dielectric constant). The height of wrinkles directly impacts the effective surface area available for contact and separation. Within a specific range of bending curvatures, a more significant variation in wrinkle height leads to substantial changes in the contact-separation distance and effective surface area. This, in turn, results in an enhanced triboelectric voltage response to bending motion. Therefore, the chosen parameter merely serves as a representation to elucidate the crucial relationship between wrinkle height variation and sensitivity, which is fundamental for the effective performance of NB-fiber as a self-powered triboelectric sensor.

8. What is the diameter of NB fiber before and after the stress being released? As it is named as "fiber", the reviewer is interested in knowing its dimension.

Response: Thank you for your inquiry regarding the dimensions of NB-fiber.

1) The diameter of NB-fiber typically falls within the range of 1.5 mm to 2.5 mm before stress is released. After stress release, the diameter varies between 0.8 mm to 2 mm, depending on specific parameters during the manufacturing process.

2) In the realm of fiber-based materials, the definition of 'fiber' encompasses more than just diameter; it involves a material having a large aspect ratio (typically exceeding 500), a small diameter (usually less than $100 \mu\text{m}$), and a certain degree of flexibility [Ref.: Adv Mater 32(5): 1902301 (2020)]. These criteria are crucial in defining what constitutes a fiber.

Within the domain of fiber electronics, an acceptable diameter range can extend to a few millimeters. When we refer to the dimensions presented in Table R4, it becomes apparent that the diameters listed there exceed 1 mm. Hence, while traditional fibers typically have smaller diameters, the acceptance of larger diameters within the context

of fiber electronics allows devices like NB-fiber to be named as fibers.

Table R4 Examples of literature on current fiber electronics

No.	Ref.	Sample Name	Sample Photo	Diameter
1	Nature 603, 616–623 (2022).	Acoustic fibre		1 mm
2	Sci. Adv. 7, eabl3742 (2021).	Zinc ion fiber battery		1 mm
3	Nat Commun 12, 3211 (2021).	Self-winding fiber actuator		3.5 mm
4	Nat Commun 12, 3317 (2021).	Digital fibres		1.2 mm
5	Nat Commun 11, 6006 (2020).	Thermoelectric fibers		0.8 mm
6	Nat Commun 11, 3537 (2020).	Triboelectric fibers		0.75 mm

7	Nat Commun 12, 1416 (2021).	Super-elastic fibers		0.8 mm
---	-----------------------------	----------------------	--	--------

9. Fig2 (b) scale bar is now shown. Fig 2 (d), What is meant by "identical stimuli"? The comparison is invalid if the test conditions are different. Please refer to comment 2. Fig 2 (e): Why curvature values are negative whereas those values in figure 2c are positive? Please add a clear description.

Response: Thank you for your meticulous review.

1) We apologize for the missing scale bar in Fig 2 (b), and we have rectified this in the updated version, as shown below:

b

Fig. 2 b The images illustrate the test setup of NB-fiber in both the initial and flexure states. scale bar: 2 cm.

2) Regarding Fig. 2 d, the term 'identical stimuli' refers to consistent bending angles applied to the sensor during testing. The testing details and data results utilized in the cited reference, as shown in Table R5, have been adopted. We have performed a comprehensive analysis, calculating and comparing the sensor's output variations under different bending conditions. Upon meticulous examination, we identified differences between the methods defined in Ref. 47 and the conventional bending sensor testing procedures.

To ensure the consistency of test conditions and maintain the validity of comparisons, we have made necessary adjustments by replacing the data from Ref. 47 with the updated ref.: Sci Adv 4, eaaq0118 (2018). Corresponding changes have been marked in

red in the revised manuscript and supplementary materials.

Table R5 Summary of testing details and data results from the cited reference

No.	Ref.	Test Details
47	Sci. Adv. 7, eabg4041 (2021).	 The graph plots $\Delta R/R_0$ (%) on the y-axis (0 to 200) against h/l_0 on the x-axis (0.0 to 2.0). A dashed green line shows a downward curve, and a solid orange line shows an upward curve. An inset schematic labeled 'Bending' shows a beam of length l_0 and height h being bent into a V-shape.
48	Nat Commun 12, 3211 (2021).	 The bar chart shows F (MΩ) on the y-axis (4 to 20) for bending degrees of 0°, 30°, 60°, and 90°. The values increase from approximately 8 MΩ at 0° to 18 MΩ at 90°. An inset image shows a hand with a sensor on the finger.
49	Adv. Mater. 28, 722-728 (2016).	 The graph plots $\Delta R/R_0$ on the y-axis (0.0 to 1.0) against Bending angle (degree) on the x-axis (0 to 60). The data points show a non-linear increase. Inset images show a hand in flexion.
50	Adv. Funct. Mater. 31, 2103703 (2021).	 The graph plots $\Delta R/R_0$ (%) on the y-axis (-30 to 0) against Bending angle (rads) on the x-axis (0 to π). The data points show a non-linear decrease. An inset diagram shows a bent beam with angle θ.
New 47	Sci. Adv.4, eaaq0118 (2018).	 The graph plots R/R_0 on the y-axis (1 to 2.2) against Finger angle (°) on the x-axis (0 to 40). Two curves are shown: 'Bending direction' (black squares) and 'Releasing direction' (red squares). An inset image shows a hand with a sensor on the finger.

New 50	Sci Adv 6, eabb4246 (2020).	✧	This work	
✧ **Our revision to the manuscript:**

We added Table R5 as Supplementary Table 2 in the revised supplementary materials. Corresponding changes have been marked in red in the revised manuscript and supplementary materials.

Fig. 2 d A sensitivity comparison was conducted between the current study and the latest findings reported in the literature, normalizing the rate of change of signal amplitude under identical stimuli.

3) The discrepancy in curvature values between Fig. 2c and Fig. 2e arises due to the different measurement setups. In Fig. 2e, the bending tests were conducted using the Electro Force 3230 equipment, where the default displacement direction is downward, with zero displacement and force at the origin. In this setup, downward displacements are represented as negative values, resulting in negative curvature values. On the other hand, in the electrical performance testing, curvature values were derived from direct measurements, considering only their magnitudes. In this context, the direction of curvature does not carry significance, as it is treated as a scalar quantity.

We acknowledge the oversight in not explicitly addressing this distinction, leading to the confusion. To rectify this, we have revised Fig. 2e to clearly indicate the downward

displacement direction, aiming for enhanced clarity.

✧ **Our revision to the manuscript:**

Fig. 2 e Flexural fatigue testing of NB-fiber.

We appreciate your attention to detail, and your feedback has greatly contributed to improving the accuracy of our representation.

10. *The conductive gel is normally maintained in its gel form to provide optimal conductivity. If the gel is dried, the conductivity will be compromised. However, it seems that the ionic gel used in the paper is cured and therefore the reviewer is interested in knowing its conductivity before and after being cured. Is flexibility of electrode impacted after gel being cured? Please provide SEM imaging of ionic gel electrode to evidence its integrity.*

Response: We appreciate the reviewer's interest in the ionogel conductivity before and after curing. It is essential to note that the ionogel used in our study was not dried. When the AAm content in the copolymer is low, the ionogel can be transparent, resembling the appearance shown in Fig. R26a. This transparency difference is not due to drying but is a result of the different solubility of PAAm in the ionic liquid [Ref.: Nat Mater 21, 359–365 (2022)]. Indeed, ionogels prepared with ionic liquids are not prone to drying, unlike water-based hydrogels. The conductivity measurements before and after curing were calculated using the provided formula $\sigma = \frac{d}{Z S}$. The conductivity value before curing was 2.36 mS/cm, while after curing, it was 2.15 mS/cm, as shown in Fig. R26b.

To address flexibility concerns, we conducted experiments where ionogel samples were exposed to ambient air for one week, maintaining good flexibility, as depicted in Fig.

R26c. To prevent damage from the ionic liquid to the scanning electron microscope, we froze the ionogel using liquid nitrogen, dried it in a freeze dryer for 72 hours, and then performed SEM imaging, as shown in Fig. R26d i. This image displays the morphology of the ionogel. For a closer examination of its skeletal structure, we subjected it to ionic exchange, followed by freeze-drying for an additional 72 hours. The resulting structure appears intact, as illustrated in Fig. R26d ii.

Fig. R26 **a** Ionogels with different contents of AAm show varying degrees of transparency. **b** The conductivity of ionogels before and after solidification. **c** Ionogel flexibility comparison approximately one week after exposure to air. **d** SEM images of ionogels after ion exchange with water and freeze drying: **i** unexchanged region **ii** fully exchanged region.

✧ **Our revision to the manuscript:**

We added Fig. R26b, d as Supplementary Fig. 6d, e in the revised supplementary materials. Corresponding changes have been marked in red in the revised manuscript and supplementary materials.

11. Page 11. “They confer nrous benefits regarding signal anti-interference, effectively minimizing signal crosstalk and ensuring the transmission of high-fidelity signals”.

Two issues: 1) typo; 2) vague statement due to missing reason.

Response: 1) Thank you for pointing out the typo, and we sincerely appreciate your feedback. The correct term is indeed 'numerous.'

✧ **Our revision to the manuscript:**

To clarify and strengthen the statement, we have revised it as follows: ‘**They confer numerous benefits regarding (high-frequency) signal anti-interference, effectively minimizing signal crosstalk and ensuring the transmission of high-fidelity signals.**’ Corresponding changes have been marked in red in the revised manuscript and supplementary materials.

2) The statement in question pertains to the benefits associated with ionogels, which **have been discussed in previous literature** [Ref. Nat Commun 9, 2630 (2018) and Adv Mater 34, 2205376 (2022)]. These benefits stem from the unique properties of ionogels, particularly in the context of signal anti-interference.

ionogels, containing ionic compounds like salts or ionic liquids, exhibit distinctive behavior due to the presence of ions. Unlike metal conductors that rely on free electrons, ionogels have ions constrained within the gel's structure, resulting in relatively slow ion mobility, typically at a sub-micron scale. This slow mobility leads to a lower propagation velocity of signals in ionogels, making them less suitable for high-frequency signal transmission. However, due to their slower response, ionogels can effectively counteract high-frequency interference, enhancing signal integrity. This characteristic helps minimize signal crosstalk and ensures the transmission of high-fidelity signals, addressing the concerns related to interference and signal quality.

Fig. R27 The micro-mechanisms of dielectric constants involve different frequency response ranges for electron and ion absorption and dissipation of electric fields. Under high-frequency electric fields, electrons typically respond more rapidly to changes in the electric field due to their lighter mass, resulting in shorter response times. In contrast, ions (usually part of atoms and molecules) have larger mass, and their response to the electric field is relatively slower. This difference in mass leads to distinct response rates between electrons and ions in high-frequency electric fields. [Ref.: Measurement 179, 109472 (2021).]

To further substantiate our conclusion, we conducted experiments involving both metal electrodes and gel electrodes under different electromagnetic interference conditions, as illustrated in Fig. R28. We employed powered appliances to simulate 50 Hz signals commonly encountered in environmental settings. The results revealed substantial disparities in signal behavior between the two electrode types. Metal electrodes exhibited discernible interference, thereby impacting the quality and stability of electrical signals. Conversely, ionogel electrodes demonstrated exceptional resilience, exhibiting minimal interference and preserving signal integrity even in the presence of electrical appliances operating at 50 Hz.

Furthermore, we scrutinized the performance of both metal electrodes and ionogel electrodes under a 20 kHz electric field. The findings indicated that ionogel electrodes surpassed metal electrodes in this high-frequency electric field, underscoring their

superior anti-interference properties. These results underscore the potential of ionogel electrodes in enhancing signal reliability and quality across diverse applications.

Fig. R28 The Anti-interference test of different electrodes on electrical signals. **a** Various test scenarios: i. No additional electromagnetic sources nearby. ii. Operation of electrical appliances at a frequency of 50 Hz. iii. 20 kHz electric field generator. **b** Performance of metal electrodes and ionogel electrodes under operation of electrical appliances. **c** Performance of metal electrodes and ionogel electrodes under a 20 kHz electric field.

✧ **Our revision to the manuscript:**

We added Fig. R28 as Supplementary Fig. 10 in the revised supplementary materials. Corresponding changes have been marked in red in the revised manuscript and supplementary materials.

12. Page 12. 1) Is buckling structure formed consistently? If so, what is the evidence; 2) If water get into the gap, does the performance degrade? This is not about washing test. It is for a scenario where user get sweating and therefore it could impact the performance of the NB fibre; 3) Does NB fiber/sensing system able to measure static pressure? Or it is limited to only dynamic pressure? If so, how could you compare with piezoresistive sensors which is for static pressure.

Response: Thank you for your inquiry and valuable feedback.

1) **Yes, the buckling structure is formed consistently.** In response to the inquiry about the consistency of the buckling structure formation, our research has undertaken a

comprehensive investigation. To assess the consistency of the NB-fiber properties, we conducted an extensive analysis on a 50-meter-long NB-fiber, as depicted in Fig. R13. The fiber's morphology was examined at various intervals, including 1 meter, 2 meters, 5 meters, 10 meters, and the entire 50-meter length.

Fig. R13 Uniformity demonstration of 50-meter continuous production process samples. a macroscopic display and b corresponding local magnified photographs.

2) Regarding the concern about the impact of water on NB-fiber performance, it's important to note that the TPU material used in NB-fiber possesses hydrophobic properties, and the nanostructure exhibits water-resistant capabilities. As shown in Fig. R29, the contact angle of the TPU nanofiber membrane can reach 141.1° , which remains at 136.9° after five minutes. This demonstrates its ability to repel water molecules effectively. Additionally, as shown in Supplementary Movie R2, even small water droplets (30 μL) have difficulty adhering to the NB-fiber surface. When the mass of the water droplet is increased, it tends to detach from the fiber surface upon slight agitation (Fig. R30a). With further increases in water droplet mass, it eventually

detaches from the NB-fiber due to gravity (Fig. R30b).

This hydrophobic surface structure prevents moisture from entering the gaps when NB-fiber comes into contact with sweaty skin in practical usage scenarios. **Hence, we suggest minimal impact of normal perspiration or rain exposure on NB-fiber performance. However, humidity changes caused by prolonged sweating or climate conditions may result in a noticeable decline in the electrical performance of NB-fiber,** as demonstrated in Fig. R23. Additionally, if subjected to prolonged submersion or exposure to a large volume of water, hydrostatic pressure could potentially facilitate water penetration into the gaps, causing a temporary reduction in performance, as depicted in Fig. R31.

Fig. R29 Contact angle test of TPU nanofiber network.

Fig. R30 The attachment of water droplets of different sizes on the NB-fiber surface. Due to the non-affinity at the interface, when the water droplet is subjected to slight disturbance or increased mass, it easily slides off from the NB-fiber surface.

Fig. R31 Triboelectric performance of NB-fiber before and after immersion in water. Performance decreases when the gap is filled with water.

3) **It's important to clarify that NB-fiber are primarily designed to measure dynamic pressure and are not suitable for measuring static pressure.** This limitation arises from the fundamental working principle of triboelectric-based sensors. Triboelectric-based sensors essentially operate as displacement sensors. They depend on the change in the electric field generated during the contact and separation process to generate a response, as illustrated by Equation $V = \frac{\sigma x(t)}{\epsilon_0}$. Here, ' σ ' represents the surface charge density, ' $x(t)$ ' denotes the friction gap, and ' ϵ_0 ' stands for the dielectric constant of the dielectric material, as shown in Fig. R32. In the case of static pressure, there is no change in the electric field over time since it does not involve dynamic motion or displacement. As a result, sensors based on the triboelectric effect are inherently designed for the measurement of dynamic pressure, where variations in the electric field correspond to changes in pressure.

To provide a clearer comparison, Table 1 in our paper is intended to illustrate the key points regarding pressure sensors based on different sensing principles and to highlight their respective advantages and limitations. It is not meant to emphasize the superiority of one type of sensor over another but rather to present a comprehensive overview of their characteristics.

Fig. R32 The theoretical model and equivalent circuit diagram of a single-electrode TENG. [Ref.: Nano Energy 14, 161-192 (2015).]

13. Page 13: It is not clear on how NB fiber is physically used (positioning/installation/wiring/etc.)? E.g. Used as a single fibre or just woven into a piece of fabric. Does pre-tension caused by the positioning/installation have impact to the reading?

Response: Thank you for your insightful comments.

1) While the positioning of the NB-fiber differs slightly between tests, fundamentally, we utilize a single fiber for pulse measurement and muscle force monitoring. For pulse measurement, a NB-fiber is wrapped multiple times around the wrist, enhancing deformation during pulsation. For muscle force monitoring, a single NB-fiber is affixed directly onto the arm protector via sewing. In both cases, we test a single NB-fiber positioned directly rather than being woven into a fabric.

Fig. R33 The placement and configuration of NB-fiber for pulse and muscle force detection are depicted. Electrodes are firmly embedded by metal wires into the ionogel, sealed with UV-curable adhesive. In this configuration, the signal input terminal of the testing device connects to the electrodes of NB-fiber, while the other terminal is linked to an isolated ground line.

2) **The pre-tension force from positioning/installation influences readings.** To address this, our software incorporates threshold adjustment windows (Fig. R34). These windows utilize signal values from a relaxed arm state immediately after donning as initial references. Our approach assesses relative changes in magnitude, making it insensitive to specific numerical values resulting from positioning or installation.

Fig. R34 Muscle force monitoring software interface, with the red box indicating the initial threshold setting area

14. Page 15: *For breathing measurement, how accurate the NB sensor is compared to the gold standard? What are you comparing with in your research? where the sensor was positioned and why?*

Response: Thank you for your valuable comments.

1) To assess the NB-fiber's accuracy in respiratory monitoring, control comparisons involving three individuals were conducted at a hospital. Vital lung function parameters, including vital capacity and respiratory rates, were measured, and compared between the NB-fiber output signal and reference pulmonary function tests. Respiratory signals from the NB-fiber were analyzed against measurements from a medical-grade monitor. Confidence interval analysis indicated error margins below 5%, as shown in Fig. R35. Respiratory rates for the two subjects are illustrated in Fig. R36, revealing superior pulmonary function in the male subject. Overall, the control comparisons demonstrated the NB-fiber's high accuracy in measuring key respiratory parameters compared to measurements conducted on medical equipment.

Fig. R35 Comparative study of vital capacity tests. **a** Vital capacity was simultaneously measured using NB-fiber and a medical pulmonary function testing device. **b** NB-fiber signal curves corresponding to different pulmonary capacities. **c** Confidence intervals for NB-fiber signals and their corresponding pulmonary function test values.

Fig. R36 Comparative study of respiratory rate testing. **a** Comparative measurement of respiratory rate using NB-fiber and a medical pulmonary function testing device simultaneously. **b** Comparison of respiratory signal curves for a 20-year-old male. **c** Comparison of respiratory signal curves for a 26-year-old female. (Left: Triboelectric testing system, Right: Pulmonary function testing system).

✧ **Our revision to the manuscript:**

We added Fig. R35 and R36 as Supplementary Fig. 21 in the revised supplementary materials. Corresponding changes have been marked in red in the revised manuscript and supplementary materials.

2) **The NB-fiber was positioned between the chest and abdominal regions.** This choice aimed to capture both thoracic and diaphragmatic breathing patterns comprehensively. Literature, as summarized in Table R6, consistently supports this placement strategy.

Table R6 Installation positions and testing data references in respiratory monitoring literature

No.	Ref.	Installation position	Raw Data
1	Nat Commun 11, 3537 (2020).		2	Sci Adv 6, eaba9624 (2020).		3	Sci Adv 6, eaay2840 (2020).		4	Matter 4, 3725– 3740 (2021).		
15. Page 17: How tight the sleeve needs to be on the arm? Surely different *tightness* will have impact to the output. Please elaborate this.

Response: Thank you for the valuable feedback. Through experimentation, we defined and gauged the relative tightness of the sleeve fitting. Specifically, we define the

relative sleeve tightness (ϵ) as the percentage difference between the arm circumference (A_0) and the sleeve's initial length (S_0) prior to fitting.

$$\epsilon = \frac{A_0 - S_0}{A_0} \times 100\%$$

Where A_0 is the arm circumference without the sleeve, and S_0 is the sleeve length when laid flat before fitting, as visually depicted in Fig. R37a.

Fig. R37b further demonstrates the impact of varying sleeve tightness on muscle deformation waveforms. Remarkably, at the 3.15% tightness level, depicted in Fig. R37c, the waveforms exhibit stability during activities such as fist clenching. This stability signifies an optimal equilibrium: the sleeve is tight enough to prevent artifacts in the data but not so constricting as to impede natural movements or cause discomfort.

Fig. R37 **a** detailed examination of the impact of sleeve tightness on the signal. **a**, the test details of sleeve tightness affecting the signal. **b** muscle deformation waveforms during fist clenching under various tightness conditions. **c** signal amplitudes corresponding to different sleeve tightness levels.

16. Page 18: *Drying in a washing machine for 30 mins is not a natural drying process!*

Response: Thank you for your keen attention to detail, and we acknowledge the discrepancy in our description. While we did employ a commercial washing machine for the washing cycles, the subsequent drying process indeed involved machine drying for a set duration followed by a brief period of natural drying for cooling and to ensure its smoothness. This adaptation was made to align with real-world conditions more closely, simulating the combination of machine drying followed by a short period of natural air drying that often occurs in practical scenarios. We thank you for highlighting this inconsistency. To provide a more accurate description of our methodology, we have revised the relevant passage in the manuscript as follows: *'To evaluate washing ability, we subjected the NB-textile to ten cycles of washing and drying in a commercial washing machine, according to the rigorous washing test (see Methods)'*. This revised description accurately reflects the washing and drying process employed in our study. We thank the reviewer for their valuable input.

17. Discussion is weak without providing evidence. It provides generic statement with vague sentences such as "high sensitivity" and "high flexibility".

Response: Thank you for your insightful feedback. We acknowledge the need for more concrete evidence in our discussion. In response, we have made extensive revisions to the manuscript, marked in red for your convenience.

We have incorporated the gauge factors to customize the flexure factor for evaluating bending sensitivity. This information is highlighted in various sections of the manuscript, including the abstract, introduction, discussion, and Supplementary Note 5. Furthermore, we have comprehensively compared our flexure factor with relevant studies in Table 1. These adjustments address the earlier absence of quantitative evidence, fostering a more robust and substantiated discussion aligned with the conventions of scientific writing. Your guidance is highly valued, and we appreciate the opportunity to enhance the quality of our manuscript.

18. In general, English and presentation need to be significantly improved.

Response: Thank you for your advice, we have made meticulous modifications to this manuscript therefore, the readers may understand our work more clearly. The corrected details are highlighted in the manuscript.

Supporting information

1. S Fig 1. Comsol simulation is missing unit in legend. 2D or 3D simulation? Boundary conditions?

Response: Thank you for pointing out the missing unit in S Fig 1. In the corrected version, we have included the unit in the legend. The simulation is performed in a 2D model.

Supplementary Fig. 1 The COMSOL Multiphysics fitting of voltage output for different tribo-gaps in the triboelectric fiber.

2) In our study of the triboelectric nanogenerator, we incorporated the following specific boundary conditions commonly employed in COMSOL electrostatic simulations:

Geometry and Materials: The simulation involved defining the geometries of TPU (Thermoplastic Polyurethane), EVA (Ethylene Vinyl Acetate), and the hydrogel electrode. Material properties such as permittivity and conductivity were assigned based on the actual properties of these materials.

Charge Distribution: Surface charge density was applied on the interfaces of TPU and EVA. TPU was assigned a negative surface charge density of $-1e-7 \text{ C/m}^2$, while EVA had a positive surface charge density of $1e-7 \text{ C/m}^2$. These values represent the charge distribution on the respective surfaces.

Distance and Gap: The simulation considered different distances between TPU and EVA layers, ranging from $50 \mu\text{m}$ to $250 \mu\text{m}$. The gap between the layers was accurately defined to analyze the electrostatic interaction across varying distances.

Dielectric Properties: The permittivity values of TPU, EVA, and the hydrogel electrode were incorporated into the simulation model. These values influence the electric field distribution within the materials.

Solver Settings: The simulation utilized appropriate solver settings, such as convergence criteria and mesh refinement, to ensure accurate and reliable results.

By incorporating these boundary conditions, we aimed to simulate the realistic electrostatic behavior within the nanogenerator components.

2. S Fig 4. Simulation or experiment.

Response: S Fig.4 represents a schematic illustration of a theoretical model based on Dr. Xu Fan's research [Ref.: J. Mech. Phys. Solids 94, 68-87 (2016)]. This model investigates the contact behavior between a core and a composite cylinder. According to Dr. Xu Fan's research, when the gap between the core and shell is small or even non-existent before structural buckling occurs, it tends to induce a stable sinusoidal axisymmetric deformation pattern. Conversely, if the gap is larger, defect sensitivity arising from this gap leads to a non-axisymmetric 'diamond-like' mode during buckling. This study emphasizes the ability to control and select instability modes (axisymmetric/diamond-like) by adjusting the gap size. The insights gained from this research have influenced the design of our fiber structures. Therefore, this model was employed to elucidate the shape recovery of shape-memory hollow tubes and control over wrinkled morphology. To provide further clarification, we have made the following modifications in the Supplementary Information.

✧ **Our revision to the manuscript:**

Supplementary Fig. 2 The illustration depicts how the interlayer gap influences and regulates fiber wrinkling amplitude in the schematic model. **a** there is no interlayer gap between the sheath and core. **b** An interlayer gap is present between the

sheath and core. This study underscores the capability to control and select unstable modes by adjusting the gap size. Specifically, when there is minimal or no gap between the core and sheath before structural buckling occurs, it results in a stable sinusoidal axisymmetric deformation. In contrast, a larger interlayer gap induces a defect-sensitive non-axisymmetric "diamond-like pattern" mode during buckling. [Ref.: J Mech Phys Solids 94, 68-87 (2016)."]

3. S fig 7. The wrinkles seem to be randomly formed so how could you control the consistency during the fabrication process? Have you measured surface roughness of NB fiber? It seems that the voltage drops significantly over 10 cm length from 0.9V to 0.3V. Practically, how could you compensate voltage drop when weaving a large piece of fabric? Comparing d and e, the voltage generated by a 5cm NB fiber is different, one is at 0.85V and the other one is at 0.6V. Why?

Response: Thank you for your inquiry regarding the apparent randomness of wrinkles in S Fig. 7 and our control measures to ensure consistency during the fabrication process.

1) Ensuring control and consistency in wrinkle formation is a crucial aspect of our research. In this context, we conducted a statistical analysis of wrinkle heights over a specific length (in this case, 2 cm) to demonstrate the stability of average wrinkle heights as an indicator of consistency during the manufacturing process. The results of this analysis, as depicted in Fig. R38, illustrate that the average height and quantity of wrinkles remain stable within the observed 2 cm length. This statistical stability indicates that while individual wrinkles may appear to form randomly, their overall characteristics remain consistent throughout the fabrication process.

Fig. R38 The wrinkle details of NB-fiber. **a** Macroscopic view of wrinkles on a 50-meter-long NB-fiber. **b** Microscopic enlargement of a randomly selected 2-centimeter-long NB-fiber (4 randomly selected sections of the fiber are labeled as i-iv). **c** Corresponding statistical analysis of wrinkle heights.

Additionally, as previously mentioned (Fig. R4), we maintain control over wrinkle

height during the manufacturing process by adjusting various factors, including pre-stretching ratios, spinning parameters, and annealing temperatures. To provide a more comprehensive understanding of how we achieve and maintain control and stability in wrinkle formation, we have included these statistical findings in our Supplementary Information, as shown in Fig. R39.

Fig. R4 The NB-fiber manufacturing process and its influencing factors.

Fig. R39 The impact of different parameters on wrinkle height. **a** Various stretching ratios. **b** Different spinning film thicknesses; **c** Diverse winding speeds of the spinning film. and **d** Statistical analysis of the influence of different annealing temperatures on wrinkle height and count.

2) Yes, we have conducted a comprehensive surface roughness measurement of the NB-fiber using the Dektak XT stylus profiler, as illustrated in Fig. R40. It is important to acknowledge that due to the limited scanning range of the equipment, our observations were confined to a relatively short length of 1 cm. Therefore, we selected two segments of the fiber from different regions for scanning. While this method may lack the precision of more advanced equipment, it allows for a relatively objective evaluation. It is noteworthy that, despite their irregularities, the wrinkles exhibited a relatively uniform and consistent amplitude.

Fig. R40 The surface roughness assessment of NB-fiber. **a** Detailed images from roughness measurements obtained using Dektak XT stylus profiler. **b-c** Surface roughness scan waveforms of NB-fiber at different locations.

✧ **Our revision to the manuscript:**

We added Fig. R38 and R40 as Supplementary Fig. 8 in the revised supplementary materials. Corresponding changes have been marked in red in the revised manuscript and supplementary materials.

3) In our study, S Fig.7d illustrates the output of different fiber lengths under the same bending angle. It is crucial to consider not only the bending angle but also the fiber length, which directly impacts the curvature, as described by the curvature formula

$$k = \frac{d\theta}{dL} = \frac{1}{R}$$

Where θ represents the angle between the curve and the tangent line, and L denotes the arc length. This formula indicates that curvature is associated with the rate of change in arc length, thereby correlating curvature with the length and bending angle of the curve, as depicted in Fig. R41.

Fig. R41 The relationship between curvature, angle, and arc length. When different lengths of arcs bend at the same angle, their curvatures vary. **Larger arc lengths correspond to larger curvature radii.**

Many previous studies have overlooked the influence of fiber length on bending performance by solely focusing on the bending angle. However, we recognize the importance of considering both curvature and length. Specifically, maintaining a consistent bending angle but increasing fiber length decreases the bending curvature. This reduced curvature leads to a smaller contact area between triboelectric surfaces, decreasing the measured triboelectric voltage. Our study emphasizes the significance of evaluating performance based on curvature rather than angle alone. As depicted in Fig. R42, we have reassessed output characteristics under varying fiber lengths while maintaining a consistent bending curvature. By holding curvature constant, we have ensured stable triboelectric voltage outputs across different fiber lengths.

Fig. R42 Triboelectric voltage of different lengths of NB-fiber under a bending curvature of 2 mm^{-1} . **a** Triboelectric voltage waveforms. **b** Error bar statistics.

We believe that assessing performance based on curvature provides a more precise representation of bending behavior. By doing so, we have addressed concerns about the performance decline over a large area during the weaving process. We hope this explanation clarifies our approach, and we appreciate your valuable feedback, which

has contributed to the improvement of our research.

4) Thank you for pointing out the difference in the generated voltage between d and e in S Fig. 7. We acknowledge that the output voltages for a 5 cm NB fiber were inherently different in these cases. In S Fig. 7d, a bending angle of approximately 100° was applied, corresponding to a curvature of about 6.2 mm^{-1} . In contrast, S Fig. 7e explores different curvatures, with 5.6 mm^{-1} corresponding to approximately 90° . This natural variation in bending amplitude directly influences the output voltage, leading to the observed differences in the generated voltage between the two cases. To avoid any confusion, we further describe S Fig. 7d and supplement Fig. R42b.

✧ **Our revision to the manuscript:**

Supplementary Fig. 11 d Triboelectric voltage of different lengths of NB-fiber under a bending curvature of 2 mm^{-1} .

4. S Fig 11. It is not clear on how many NB fibers are used at each measurement position. In a) four data points are shown but in b), only one NB fibre?

Response: Thank you for pointing this out. Upon re-examination, we acknowledge the confusion around the number of NB-fibers used at each measurement position in Supplementary Fig. 11.

In S Fig. 11a, the four data points at each location represent repeat measurements from a single NB-fiber. Due to minor variations in grip force between trials, small differences in triboelectric output were observed even when measuring in the same spot. S Fig. 11b indicates the distinct positions on the arm where a single NB-fiber was installed for

testing. Each position directly corresponds to the data presented in S Fig. 11a. To eliminate ambiguity, we have updated S Fig. 11a in the supplement to explicitly show that only one NB-fiber was installed sequentially at different locations. Error bars have also been added to represent the natural variations between repeated trials from a solitary sensor.

✧ **Our revision to the manuscript:**

Supplementary Fig. 18 Different forearm positions for muscle force testing. a Muscle deformation monitoring at various locations such as dorsum of the hand, wrist, forearm, and elbow, under the same grip force. **b** Placement of NB-fiber during the testing.

5. S Fig 12. Again, it is not clear on how many NB fibrers are used? If more than one, how do they electronically connected?

Response: Thank you for your careful review.

1) For S Fig. 12, we were monitoring muscle deformations corresponding to different grip strengths. Hence, only one NB fiber was used for monitoring. The primary muscle monitored during fist-gripping and lifting activities was the flexor carpi radialis muscle, located on the inner side of the forearm, as depicted in Fig. R43. The electrodes of the NB-fiber were connected to Keithley 6514E for measurements. The arm guard in the illustration is constructed with NB-fibers arranged both longitudinally and transversely. However, subsequent tests revealed that axially placed NB-fibers along the arm provided more distinct results in monitoring various muscle exertions, aligning with the natural distribution pattern of muscles. Therefore, the transverse NB-fibers did not play a role in this context. To enhance clarity, we have supplemented the installation positions of the NB-fibers, as depicted in Fig. R43b.

Fig. R43 In the monitoring of Flexor carpi radialis muscle deformation under different gripping forces, the installation details of the NB-fiber are as follows. **a** The Flexor carpi radialis muscle is located at the middle position on the inner side of the forearm. **b** The installation position of the NB-fiber in the sleeve. **c** The circuit board and Bluetooth module used for single-channel signal acquisition.

❖ **Our revision to the manuscript:**

Supplementary Fig. 19 Monitoring flexor carpi radialis muscle deformation under different grip forces. **a** Muscle deformation signal observed during loose grip. **b** Muscle deformation signal recorded while lifting a 5 kg weight. **c Detailed description of the testing procedure employed during the experimental process.**

2) In addition, regarding the query 'If more than one, how do they electronically connected?' we would like to provide further clarification. During different motion recognition tasks, we monitored the exertion of various muscles using a total of 8 NB-fibers distributed at different positions on the forearm. These fibers were connected to a multi-channel sensor array circuit board, as depicted in Fig. R44. The triboelectric signal modulation circuit includes a charge amplification circuit, a 50 Hz notch filter circuit, a voltage boosting circuit, and a post-amplification circuit, as illustrated in S

Fig. 26. The mention of "12 channels" indicates that the system has the capacity to handle signals from up to 12 different sources simultaneously, rather than being limited to only 12 channels working at the same time. This flexibility allows for versatile data collection and analysis, enabling a comprehensive understanding of various signal sources and their interactions. Thank you for your understanding and valuable feedback, which have significantly improved the precision of our work.

Fig. R44 **a** Connection details of multi-channel sensor signal acquisition. **b** Multi-channel sensor signal acquisition and Bluetooth transmission circuit board.

Supplementary Fig. 26 Circuitry and functional analysis of the multi-channel sensing system. **a** Physical layout of the 12-channel circuit board. **b** Various functional modules integrated into the circuit board. **c** Sensing signal processing circuit.

6. *S Fig 13. Repeatability of gesture recognition?*

Response: Thank you for your insightful query regarding the repeatability of gesture recognition. Based on the methodology outlined in literature [Ref.: IEEE Int. Conf. Consum. Electron. 2019: 1-4. doi: 10.1109/ICCE.2019.8662078.], we conducted a meticulous evaluation by designing a series of standardized gestures (12° , 18° , 29° , 40° , 50°) and subjected them to multiple tests under consistent environmental conditions. The data collected from these tests underwent rigorous analysis. By utilizing variance analysis, as illustrated in Fig. R45, we precisely assessed the extent of variation in gesture data across multiple repetitions. This robust approach, detailed in the figure, ensured the repeatability of our gesture recognition system. Your attention to this critical aspect of our study has been invaluable.

Fig. R45 **a** Reproducibility of gesture sensor signal waveforms at various bending angles. **b** Statistical analysis of amplitude errors in gesture sensor signal waveforms at different bending angles.

Thank you again for your valuable comments and suggestions.

Response to Reviewer #3

The manuscript introduces a groundbreaking technique for fabricating micro-flexure-sensitive electronic fibers using nanofiber buckling. The approach results in high-specific surface tribo-interfaces and enhanced sensitivity, making the fibers suitable for a range of applications. The authors also demonstrate the versatility of the technology through custom software applications, demonstrating its potential in fields such as acupuncture mapping, muscle group force monitoring, and wearable devices for rehabilitation and physical therapy. Furthermore, the manuscript discusses the application of the technology in prosthetics and exoskeletons, highlighting its potential to revolutionize healthcare and wellness. Despite the interesting results, the manuscript raises several major questions and concerns that, if not addressed, might hinder progression to the next stage. Below, I have outlined my comments for the authors' consideration:

Response: Thank you for the positive feedback. We have carefully revised the manuscript according to your comments. The replies to each of your concern are listed below.

1. *Could the authors provide a more detailed explanation of how the ionogel electrode served as an electrically conductive electrode? Detailed information on the measurement setup would greatly enhance the reader's understanding of this aspect.*

Response: Thank you for the valuable feedback. Addressing the question regarding ionogel electrodes, it is essential to delve into the fundamental principle of TENGs.

1) TENGs harness the phenomenon of charge separation that occurs when materials with different electron affinities come into contact and then separate. This process exploits the electrostatic induction effect, enabling the conversion of mechanical energy into electrical energy. Initially, upon contact, electrons move from the material with lower electron affinity (tribo-positive) to the material with higher electron affinity (tribo-negative). This results in localized charge accumulation, as depicted in Fig. R46a. Upon separation, opposing charges are induced on the electrode surface, initiating electron flow and generating an electric current, as shown in Fig. R46b. During the re-contact phase, the dielectric's charge shielding properties cause a reversal in electron movement, leading to the production of a reverse current, as demonstrated in Fig. R46c.

The electrostatic induction effect plays a crucial role in releasing static field energy, with the electrode configuration demonstrating the ingenious design principle.

Fig. R46. Working mechanism of the single-electrode TENG based on vertical contact separation. [Ref.: "Triboelectric Nanogenerator: Single-Electrode Mode." Triboelectric Nanogenerators. Springer International Publishing, 2016, 91-107.]

Ionogel electrodes, similar to traditional metal electrodes, conduct electric current through electrostatic induction. This ability arises from their ionic conductivity and the creation of electric charge gradients during contact and separation. When an ionogel electrode is grounded via a conductor, the contact-separation generates an electric field, causing induced potential to accumulate on the ionogel electrode. Electrons migrate towards the grounded point due to the electrostatic potential difference, while ions within the ionogel redistribute to maintain electrical neutrality, as shown in Fig. R47c. This redistribution ensures a balanced charge transfer, resulting in an electric current flowing through the grounded connection.

In equivalent circuits, ionogels are represented as a combination of resistive and capacitive components due to their ionic conductivity, giving them capacitive-like behavior. This representation accurately mirrors their electrical properties, as depicted in Fig. R47b. Moreover, ionogels feature low interfacial impedance, enhancing charge transfer efficiency in diverse applications. **Compared to metal electrodes, ionogel electrodes offer distinct advantages, including exceptional mechanical flexibility, low-impedance interfaces for efficient charge transfer, effective electromagnetic shielding, and responsiveness to electrostatic induction.** These attributes render ionogel electrodes highly versatile and well-suited for a wide range of sensing applications and beyond.

Fig. R47 Equivalent circuits of metal electrodes and gel electrodes, along with a schematic representation of ionogel charge redistribution.

2) In our experiments, ionogel electrodes were encapsulated with stainless steel wire and UV-curable adhesive, as shown in Fig. R48. Triboelectric performance was tested using a Keithley 6514E electrometer. One terminal connected to the stainless steel wire, the other to the ground, ensuring efficient charge transfer.

Fig. R48 Detailed testing photos and schematic of the wiring method for NB-fiber.

2. Although the manuscript presents measurements of the output voltage, similar to the first comment, it is necessary to describe the electrical characteristics of the device under specific electrical load conditions or impedance environments. While the generator's performance was evaluated at several Mega Ohms, information about the analysis conducted on the fiber-type device under different loads is crucial for comprehending its overall performance.

Response: Thank you for your valuable feedback.

1) In our experiments, all electrical performance tests were directly connected to the Keithley 6514E electrometer without additional resistors, unless specified otherwise. During testing for triboelectric signals, the load conditions were adjusted based on the

voltage range setting of the electrostatic voltmeter (as per the Model 6514 System Electrometer Instruction Manual, the 2 V range corresponds to a load of 100 GΩ). Recognizing the reviewer's point about the influence of varying load conditions on the sensor device's electrical performance, we conducted additional tests measuring triboelectric voltage outputs under diverse load conditions. The results are depicted in Fig. R49, providing a comprehensive evaluation of the NB-fiber's overall performance.

2) To address the influence of load conditions on the sensor device's electrical performance, we performed tests measuring triboelectric voltage outputs across resistances varying from 10 MΩ to 100 GΩ, as illustrated in Fig. R48. Fig. R48b presents the triboelectric voltage response of NB-fiber under different resistances, accompanied by corresponding peak power calculations using the formula $P = \frac{U^2}{R}$. The graph clearly shows that the triboelectric voltage increases with higher loads, **reaching its peak output at approximately 5 GΩ.**

Fig. R49 **a** Wiring methods under different loads and **b** corresponding triboelectric voltage and peak power of NB-fiber.

3. In Figure S8, mechanical stress characteristics are provided concerning angle and repetition count, but it would be desirable to include the corresponding electrical

properties as well. Despite the mention in the manuscript, the actual data is missing. At the very least, a discussion or perspective on the electrical properties should be included.

Response: Thank you for your valuable feedback. We appreciate your insightful suggestion, and in response, we have incorporated electrical property data corresponding to mechanical stress characteristics, specifically focusing on 10,000 bending cycles. In Fig. R50, the graph illustrates the stable performance of the NB-fiber throughout 10,000 bending cycles, indicating its exceptional durability and consistent electrical output.

Fig. R50 Electrical properties of NB-fiber in 10,000 bending fatigue tests.

Acknowledging the potential impact of ionogel electrode fatigue on conductivity, we monitored the device's conductivity during 10,000 bending cycles. The graph in Fig. R51 displays the conductivity of the NB-fiber after 1, 10, 100, 1000, and 10,000 bending cycles, respectively. These results clearly indicate the sustained stability of the entire device even after extensive mechanical stress, affirming the robustness of our design.

Fig. R51 The conductivity of the NB-fiber after 1, 10, 100, 1000, and 10,000 bending cycles.

✧ **Our revision to the manuscript:**

We added Fig. R50 and R51 as Supplementary Fig. 15 in the revised supplementary materials. Corresponding changes have been marked in red in the revised manuscript and supplementary materials.

4. The rationale for utilizing FEP as an external substance for washing after tests raises curiosity. It would be valuable to discuss the test results stemming from friction between the original core and shell materials. A discussion by the authors on this topic is warranted. Assuming successful encapsulation, minimal changes in characteristics before and after washing are predictable. More meaningful data would involve the frictional electrical properties between core and shell materials.

Response: We deeply appreciate the insightful feedback provided by the reviewer, which encouraged us to conduct a more in-depth analysis of the frictional characteristics between the core material (EVA) and the shell material (TPU nanofibers) in our study. To simulate the frictional forces exerted by washing machines, we conducted the Martin-dale abrasion test. In this test, a load of 260 grams was applied to the samples, with the TPU nanofilm clamped on the upper abrasion head and a layer of EVA nanofiber membrane placed beneath on the lower abrasive disc, as depicted in Fig. R52a. After 10,000 cycles, the outer shell material exhibited no discernible signs of wear, as depicted in Fig. R52b. SEM images detailing the surface morphology of the NB-fibers before and after friction are presented in Fig. R52c. Post-friction (ii), the samples displayed fiber aggregation and increased roughness due to enhanced charge accumulation resulting from friction, leading to fiber surface aggregation.

Importantly, this phenomenon did not significantly impact the electrical performance output of the frictional device, as demonstrated in Fig. R53. The triboelectric voltage under different Martin-dale abrasion cycles did not exhibit a noticeable decrease, highlighting the stability of the system. Moreover, as discussed in Comment 3 (from Fig. R50 to Fig. R51), we monitored the NB-fiber's triboelectric output and conductivity changes over 10,000 bending cycles, confirming the robustness of its electrical properties during prolonged cycling.

Fig. R52 The Martin-Dale Abrasion Test. **a** The Martin-Dale Abrasion Tester subjected the sample to multiple friction cycles with a 260g weight applied to the standard friction sample. **b** Morphology of the test sample after various friction cycles. **c** SEM images of the sample before and after friction cycles, i represents pre-friction, ii represents post-10000 cycles of friction.

Fig. R53 The triboelectric voltage of shell-core materials after different cycles of the Martin-Dale Abrasion Test.

Additionally, we evaluated the NB-fiber's stability before and after 1 to 10 wash cycles, as depicted in Fig. R54, thereby enhancing the assessment of its washability performance. The provided data reaffirms NB-fiber's resilience over extended bending cycles and its stability after multiple washing cycles, reinforcing its suitability for diverse applications.

Fig. R54 The triboelectric performance test of NB-fiber after 10 water washes, with a length of 10 cm and a bending angle of 30° .

✧ **Our revision to the manuscript:**

We added Fig. R54 as Supplementary Fig. 25d in the revised supplementary materials. Corresponding changes have been marked in red in the revised manuscript and supplementary materials.

5. The values for "robustness" in Table 1 require explanation regarding their derivation and units.

Response: Thank you for your valuable input. We acknowledge the need for clarification regarding the term 'robustness' in Table 1, including its derivation and units. **In the context of our study, 'robustness' refers to the cyclic lifespan or durability of the tested components.** It signifies the NB-fiber's ability to withstand repeated cyclic loading or usage without significant degradation in performance. This measure is crucial in assessing the practical applicability and longevity of the NB-fiber in real-world scenarios. To offer a more precise explanation and address this concern, we will revise the manuscript to incorporate a detailed description of how the term 'robustness' is determined, along with its measurement units, as follows: 'Note: Robustness in this table is assessed based on cycle life (unit: 1)'.

Furthermore, our survey of relevant literature on 'robustness' metrics in electronic sensing devices revealed a consistent trend. Studies, as illustrated in Table R7, universally utilize cycle life as a measure for robustness evaluation. This includes assessing the retention of initial performance within a range of xx%, a widely acknowledged criterion with variations in specific thresholds across different studies. We genuinely appreciate the reviewer's diligence in improving the clarity and comprehensibility of our manuscript.

Table R7 Literature compilation on robustness evaluation in the current field of flexible electronics using cycle counts

No.	Ref.	Raw Data	Original Text
1	Nat Commun 14, 2323 (2023).	 Normalized PCE Bending Cycles (Counts) Legend: - PM6:PY-V-γ - PM6:PY-V-γ:PCBM - PM6:PY-V-γ:PFBO-C12 Bending radius = 3 mm	Device stability and mechanical robustness.

2	Nat Commun 14, 4902 (2023).		Thermomechanical robustness of F-DLC.
3	Adv Mater 35, 2211342 (2023).		The performance remains the same after multiple stretch recovery cycles, demonstrating the robustness of the PIE9
4	Nat Commun 13, 2190 (2022).		•••textiles with comparable mechanical robustness (>10 ⁴ bending cycles)
5	Adv Mater 33, 2103755 (2021).		The excellent elasticity and robustness of Iono LCE fiber render the sensor with outstanding durability.
6	Nat Commun 12, 3211 (2021).		•••over 1270000 cycles without obvious fatigue, exhibiting high robustness.
7	Nat Commun 11, 3530 (2020).		•••and long-term durability and repeatability for more than 5000 cycles.

6. How do the mechanical and electrical characteristics change with varying temperatures? Given the nature of shape memory polymers, alterations in crystal characteristics with temperature shifts should be addressed. Results or discussions on this matter are needed.

Response: Thank you for your valuable feedback. We have conducted detailed analyses on the mechanical and electrical properties at varying temperatures to specifically address the alterations in crystal characteristics due to temperature shifts.

In our study, **the deformation temperature of the shape memory material starts at 80°C**, ensuring stability within the typical range of human activities. We extensively characterized the material's modulus and strain at various temperatures. Fig. R55 illustrates significant deformation occurring around 108°C, reaching a deformation rate of 4000%.

Fig. R55 Strain and modulus variations of shape memory polymers (EVA) at different temperatures. **a** Thermomechanical analysis of EVA demonstrates deformation initiation at 80°C. **b** Dynamic mechanical analysis of the material's viscoelastic behavior, including storage modulus, loss modulus, and tan delta values.

Furthermore, **our investigation into the influence of temperature on the electrical properties of the NB-fiber**, as presented in Fig. R56, reveals a significant fluctuation of approximately 62% in electrical performance. This substantial variation is attributed to the thermal motion of ions within the material, impacting the charge mobility and consequently the electrical output. It is crucial to note that this level of fluctuation underscores the sensitivity of NB-fiber to temperature changes. Hence, when utilizing NB-fiber in practical applications, careful attention to temperature stability is paramount to ensure consistent and reliable performance.

Fig. R56 The triboelectric voltage of NB-fiber at different temperatures.

✧ **Our revision to the manuscript:**

We added Fig. R55 and R56 as Supplementary Fig. 9 in the revised supplementary materials. Corresponding changes have been marked in red in the revised manuscript and supplementary materials.

7. The absence of a description for Figure 3g is noted. The meanings behind "empty cup" and "full cup" are absent, and quantitative information is essential. The occurrence of significant differences between empty cup, full cup, and the 5 kg case must be described. The authors should also elaborate on the potential implications of these findings.

Response: Thank you for your valuable feedback. Fig. 3g illustrates the triboelectric signal response under varying levels of grip strength exerted through lifting different weights. Specifically: "Loose grip" represents gripping an object without lifting. "Empty cup" depicts a weak grip lifting a 300 g cup. "Full cup" corresponds to a moderate grip lifting an 800 g cup of water. "5 kg case" depicts a firm grip exhibiting strength to lift a heavy 5 kg weight.

These gripping actions mimic real-world tasks requiring varying muscle exertion. Their significance in potential muscle rehabilitation applications is demonstrated in Fig. R57. For example, recovering patients may only be able to make a fist ("loose grip") or lift a light object ("empty cup") due to weak muscles. However, assessing their exertion

levels when gripping and lifting various weights provides clinically valuable insight into recovery progress. Through quantitative monitoring via NB-fiber (Supplementary Fig. 19), therapists can tailor targeted exercises accordingly. This optimization facilitates improved rehabilitation outcomes.

Fig. R57 A scene with a muscle atrophy patient attempting to pick up a glass of water, using their muscle strength to assess the likelihood of success.

Supplementary Fig. 18 Monitoring flexor carpi radialis muscle deformation under different grip forces. **a** Muscle deformation signal observed during loose grip. **b** Muscle deformation signal recorded while lifting a 5 kg weight. **c** Detailed description of the testing procedure employed during the experimental process.

✧ **Our revision to the manuscript:**

To ensure clarity and precision in conveying the main idea, we converted the weights corresponding to the objects to indicate the magnitude of grip strength. We added Fig. R57 as Supplementary Fig. 19a in the revised supplementary materials. Corresponding changes have been marked in red in the revised manuscript.

Fig. 3 g The electrical signal amplitudes corresponding to muscle strength were measured under different motions.

8. A careful review for typographical errors is warranted. For instance, "Figure S6 a) covalent ceoss links" appears to contain a typographical error.

Response: Your diligent review and sharp attention to typographical errors are greatly appreciated. Upon careful examination, we identified and promptly addressed the typographical error within the manuscript. Specifically, the text 'Figure S6 a) covalent ceoss links' has been accurately corrected to 'Figure S6 a) covalent cross-links.' We highly value your contribution, which significantly enhances the clarity and precision of our manuscript.

✧ **Our revision to the manuscript:**

Supplementary Fig. 6 Fabrication principles and performance characterization of ionogel electrodes. d Electrical conductivity before and after the solidification of the ionogel. **e** Scanning electron microscope images of the surface morphology of the ionogel (i) and the polymer skeleton (ii).

Thank you again for your valuable comments and suggestions.

REVIEWERS' COMMENTS

Reviewer #1 (Remarks to the Author):

The authors' responses demonstrate a thorough understanding of their work and a commitment to addressing the reviewers' concerns in a comprehensive and detailed manner. The additional experiments, clarifications, and revisions suggested in their responses are likely to significantly improve the quality and clarity of the manuscript. This paper can be considered for publication.

Reviewer #2 (Remarks to the Author):

The authors have comprehensively addressed the comments of the reviewer. The details of the response are of high standards. Additional details are provided and included as supplementary documents that allow the reader to get much more insight into the research.

The amended paper has not presented a high-quality of original research and novelty. It is believed that the research work will have a significant contribution to the field of wearable triboelectric sensors.

In summary, after reviewing the paper and its supplementary documents, I am happy to recommend the paper to be accepted.

Reviewer #3 (Remarks to the Author):

Thank you for submitting the revised manuscript and the detailed response letter. I have carefully reviewed both and appreciate the authors' efforts in addressing the concerns raised during the first round of review. It is noted that the authors have provided responses to most of the comments raised, and in my second set of comments, I provide additional feedback to ensure the manuscript meets the standards of the journal.

1. Have authors discussed varying electrical characteristics of Ionogel depending on ratio between monomers, AAm and AA? It may be informative for potential readers to get an idea how Ionogels, synthesized at a particular concentration, could be effectively used in diverse scenarios or alternative applications.

2. From Fig. R49, are voltage and peak power characterized under what kinds of mechanical stress? e.g., bending, stretching (if so, how much level of either N, %,...)

Responses to the Reviewers' Comments

Point-by-point responses to the reviewers' comments

We sincerely thank the reviewers for their careful and thorough review, which are indeed very helpful to make the paper more solid and smooth. We have revised our manuscript very carefully in the light of their suggestions and comments.

The following responses have been prepared to address all of the reviewers' comments in a point-by-point fashion. (Comments in black, responses in blue, changes highlighted in red):

Response to Reviewer #3

***General comment:** Thank you for submitting the revised manuscript and the detailed response letter. I have carefully reviewed both and appreciate the authors' efforts in addressing the concerns raised during the first round of review. It is noted that the authors have provided responses to most of the comments raised, and in my second set of comments, I provide additional feedback to ensure the manuscript meets the standards of the journal.*

Response: Thank you for your review and we appreciate your feedback. We have carefully revised the manuscript according to your comments. The replies to each of your concern are listed below.

1. Have authors discussed varying electrical characteristics of Ionogel depending on ratio between monomers, AAm and AA? It may be informative for potential readers to get an idea how Ionogels, synthesized at a particular concentration, could be effectively used in diverse scenarios or alternative applications.

Response: Thank you for your insightful suggestions. As illustrated in Fig. R1, the electrical conductivity decreases with an increase in AAm monomer content. The pure PAA gel exhibits a conductivity of approximately 4.25 mS cm^{-1} , while the conductivity

of pure PAAm is only 0.78 mS cm^{-1} . This decline in conductivity is attributed to the relatively poor solubility of PAAm compared to PAA in the ionic liquid [Ref.: *Nat Mater* 21, 359–365 (2022)]. The different solubilities significantly impact the ion migration within the Ionogel structure.

In Fig. R1c, the relationship between impedance phase angle and the logarithm of the modulus concerning frequency is described, revealing changes in the electrochemical interface. With an increase in the AAm ratio, impedance values exhibit an upward trend, indicating an increase in Ionogel electrochemical impedance. This rise suggests a reduction in ion mobility and decreased charge transfer efficiency at the electrochemical interface, resulting in the observed decline in conductivity. Impedance values at different frequencies are detailed in Table R1. Additionally, we analyzed the stability of gel resistance during bending (30 mm long Ionogel deflecting 5 mm towards the center), as shown in Fig. R1d. It is evident that the resistance values of various Ionogel compositions exhibit minimal changes during the bending deformation, mitigating signal crosstalk induced by pressure.

Considering these findings, Ionogels with higher electrical conductivity, such as those with lower AAm monomer content, are well-suited for applications requiring efficient ion mobility. This includes scenarios where rapid charge transfer is crucial, such as in high-performance energy storage devices or sensors. On the other hand, Ionogels with lower electrical conductivity, such as those with higher AAm monomer content, may find utility in applications where slower ion migration is acceptable, such as in certain biomedical devices or flexible electronics. We have integrated this discussion into the revised manuscript, aiming to provide a comprehensive understanding of the potential applications of Ionogels synthesized at specific concentrations.

Fig. R1 The electrical properties of Ionogels with varying AAm content. a Nyquist impedance plots for Ionogels at different AAm ratios. **b** Statistical analysis of electrical conductivity. **c** Bode plots. **d** Resistance testing during dynamic bending processes.

Table R1 Ionogel electrical characteristics with varying AAm content.

AAm (%)	Conductivity (mS cm ⁻¹)	Frequency (Hz) / log (Z /Ohm) & Phase(Z)/deg									
		0.1		1		10		100		1000	
0	4.25	4.70	76.85	3.86	76.75	3.00	78.03	2.13	76.89	1.27	74.43
10	3.88	4.72	72.27	3.94	72.05	3.11	77.15	2.23	78.33	1.35	76.99
20	3.84	4.74	70.82	3.95	71.76	3.11	77.04	2.24	78.30	1.35	76.92
30	3.45	4.63	73.83	3.87	70.86	3.10	74.74	2.19	76.55	1.32	75.63
40	3.15	4.65	66.80	3.93	70.98	3.04	75.26	2.27	75.43	1.42	73.99
50	2.95	4.74	71.29	3.93	72.40	3.12	74.75	2.27	77.65	1.38	77.11
60	2.77	4.76	71.14	3.98	69.94	3.17	74.63	2.32	77.95	1.43	76.84

70	2.37	4.81	71.98	3.99	72.73	3.18	74.10	2.34	76.22	1.47	75.43
80	1.74	4.83	60.75	4.14	66.92	3.35	73.15	2.51	77.11	1.62	75.94
90	0.85	4.88	55.32	4.27	60.08	3.51	72.26	2.67	77.11	1.79	70.63
100	0.78	4.97	53.87	4.35	61.07	3.61	69.75	2.79	74.79	1.94	70.79

✧ **Our revision to the manuscript:**

We added Fig. R1 as Supplementary Fig. 7 in the revised supplementary materials. Corresponding changes have been marked in red in the revised manuscript and supplementary materials.

2. *From Fig. R49, are voltage and peak power characterized under what kinds of mechanical stress? e.g., bending, stretching (if so, how much level of either N, %, ...).*

Response: Thank you for your insightful review comments. In Fig. R49, the voltage and peak power were characterized under mechanical stress induced by a bending motion. The experimental setup involved clamping a 50 mm long NB-fiber with a linear motor, displacing the fiber towards the center by 15 mm. This bending motion was precisely replicated using an electronic universal material testing machine, as illustrated in Fig. R2. During this process, the 50 mm NB-fiber underwent deflection, experiencing a compressive displacement of approximately 15 mm, resulting in a force of around 0.8 N. In our revised manuscript, we have incorporated this additional information into the section discussing the electrical performance tests. This inclusion ensures a more comprehensive description of the mechanical conditions during the electrical testing procedures. We appreciate your guidance in refining the manuscript for improved clarity and completeness.

Fig.R2 Characterization of the mechanical stress conditions during the electrical testing of NB-fiber.

✧ **Our revision to the manuscript:**

We added Fig. R2 as Supplementary Fig. 16e in the revised supplementary materials. Corresponding changes have been marked in red in the revised manuscript and supplementary materials.

Thank you again for your valuable comments and suggestions.